# Differential learning kinetics govern the transition from memorization to generalization during in-context learning

**Alex Nguyen**
Princeton Neuroscience Institute
Princeton University
qanguyen@princeton.edu

**Gautam Reddy**
Department of Physics
Princeton University
greddy@princeton.edu

## Abstract

Transformers exhibit in-context learning (ICL): the ability to use novel information presented in the context without additional weight updates. Recent work shows that ICL emerges when models are trained on a sufficiently diverse set of tasks and the transition from memorization to generalization is sharp with increasing task diversity. One interpretation is that a network's limited capacity to memorize favors generalization. Here, we examine the mechanistic underpinnings of this transition using a small transformer applied to a synthetic ICL task. Using theory and experiment, we show that the sub-circuits that memorize and generalize can be viewed as largely independent. The relative *rates* at which these sub-circuits learn explains the transition from memorization to generalization, rather than capacity constraints. We uncover a memorization scaling law, which determines the task diversity threshold at which the network generalizes. The theory quantitatively explains a variety of other ICL-related phenomena, including the long-tailed distribution of when ICL is acquired, the bimodal behavior of solutions close to the task diversity threshold, the influence of contextual and data distributional statistics on ICL, and the transient nature of ICL.

## 1 Introduction

Large transformer models trained to predict the next token exhibit powerful generalization capabilities. One signature of such generalization capabilities is in-context learning (ICL): the ability to solve a task based on new information presented in the context without additional weight updates (Brown et al. (2020); Dai et al. (2022); Dong et al. (2022); Garg et al. (2022); Xie et al. (2021); Olsson et al. (2022)). Arguably, the ability to interpret novel inputs on-the-fly is a core feature of any intelligent system. However, updating synaptic weights on rapid behavioral timescales is challenging, both for natural and artificial systems. The emergence of ICL in large language models (LLMs) shows that finding network states that learn on-the-fly is indeed possible. Understanding how ICL emerges in LLMs promises insights into how such algorithms may be implemented in the brain and how the data distribution, training objective and network architecture interact to enable ICL acquisition at scale.

Various methods have been used to probe the ICL capabilities of LLMs (Brown et al. (2020); Dong et al. (2022); Pan (2023); Min et al. (2022); Olsson et al. (2022)). A common ICL paradigm is to present exemplars as a sequence of item-label pairs, and measure the network's response to a target item (Chan et al. (2022); Kirsch et al. (2022); Garg et al. (2022); Akyürek et al. (2022); Von Oswald et al. (2023); Raventós et al. (2023); Bai et al. (2023)). While LLMs display remarkable capabilities on such ICL tasks, interpreting the underlying network mechanisms that give rise to these capabilities remains challenging (but see Wang et al. (2022)). Recent work has approached this challenge by examining how small transformer models solve synthetic ICL tasks (Reddy (2023); Bietti et al. (2024); Akyürek et al. (2022); Ahn et al. (2023); Von Oswald et al. (2023); Edelman et al. (2024)). We highlight two notable aspects of ICL phenomenology relevant for our current work: the influence of task diversity on whether the network memorizes a finite dataset or acquires ICL (i.e., generalizes), and how ICL is acquired (and lost) during training.

First, data distributional properties (such as task diversity and their rank-frequency distribution) influence whether the network acquires ICL or encodes the response to queries seen during training within its weights (Kirsch et al. (2022); Chan et al. (2022); Raventós et al. (2023). Following previous work, we refer to such memorization as in-weights learning (IWL). Notably, the transition from memorization (IWL) to generalization (ICL) is sharp with respect to task diversity. A curious feature of this transition is that solutions close to the task diversity threshold are bimodal (Kirsch et al. (2022)). That is, across different random number seeds, solutions either show IWL or acquire ICL but intermediate solutions are unlikely.

Second, ICL is often implemented by multi-layer computations involving nonlinear attention heads and MLPs (Olsson et al. (2022); Von Oswald et al. (2023); Reddy (2023); Bietti et al. (2024)). The rugged loss landscape induced by such multi-layer, nonlinear operations leads to long plateaus followed by a sharp drop in loss (Reddy (2023)). Finally, ICL is seemingly transient, i.e., the network gradually loses the ICL capability if it is trained for sufficiently long (Singh et al. (2023)).

It is unclear what leads to the transition from IWL to ICL acquisition with increasing task diversity. The loss is minimized when a finite training dataset is perfectly memorized. Thus, a naive hypothesis would suggest that as the task diversity increases, the network's limited capacity to memorize favors the ICL solution (Figure 1a). Our goal is to test this hypothesis by deriving a precise quantitative description of the transition, and thus we set out to identify a minimal setting that captures the phenomenon.

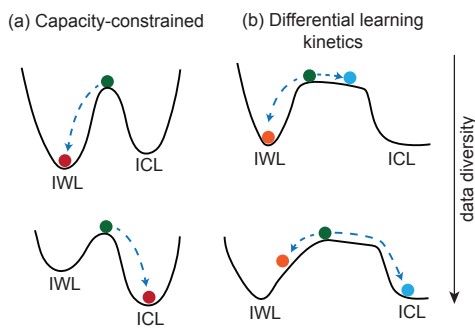

Figure 1: (a) In the capacity-constrained model, the network's limited capacity to memorize favors ICL acquisition with increasing task diversity. (b) In the differential learning kinetics model, independent sub-circuits contribute towards IWL and ICL. IWL is slower for greater task diversity. The network acquires ICL before the network can significantly memorize the training set. IWL is significantly slowed down as ICL explains most of the loss, but does eventually memorize the training set. The network subsequently loses the ICL capability due to regularization.

**Contributions and outline.** We first identify a one-layer transformer model trained on an in-context classification task that recapitulates the sharp transition from memorization to generalization. Despite its simplicity, the one-layer model displays surprisingly rich phenomenology, including abrupt ICL learning dynamics, transience and bimodal solutions close to the task diversity threshold.

Next, we derive an analytical framework that quantitatively characterizes ICL acquisition and its competition with IWL. We show that the transition from IWL to ICL for our network is governed by a *dynamical competition* between memorization and generalization (Figure 1b). However, whether the network is capacity-constrained or rate-determined depends on the network architecture, and we derive a quantitative measure to determine which of these constraints is at play. The theory predicts that the number of iterations before ICL is acquired is exponentially sensitive to the initial parameters, which in turn explains the bimodal behavior of solutions close to the task diversity threshold. The task diversity threshold follows a power law whose exponent has a non-trivial relationship with another novel memorization scaling law. ICL transience naturally follows from the theory under standard $L_2$ regularization. Finally, we validate our theory by empirically verifying these predictions using our transformer model.

## 2  TASK FORMULATION

We consider a simplified version of an ICL task proposed by Chan et al. (2022), which allows for disentangling ICL and IWL performance (Figure 2a). Before training, we generate a dataset $\mathcal{D}$ that contains $K$ item-label pairs, $\mathcal{D} = \{(x_1, \ell_1), (x_2, \ell_2), \ldots, (x_K, \ell_K)\}$. Each item $x_i$ is a $D$-

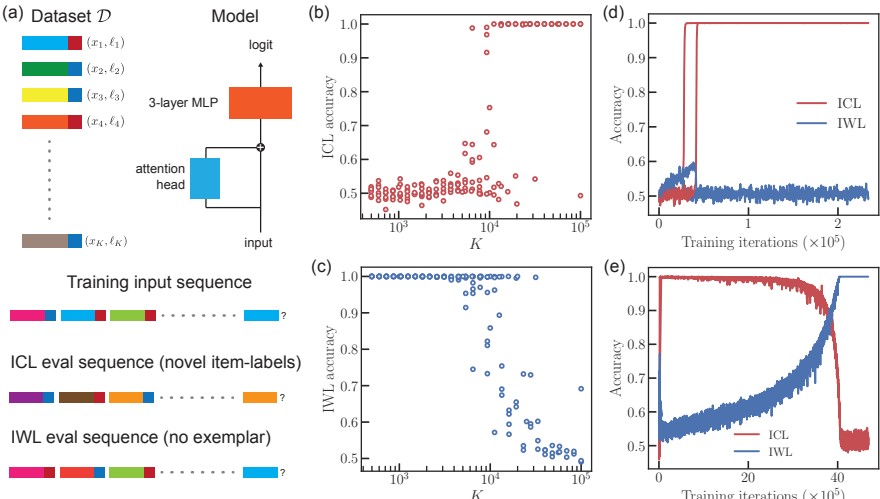

Figure 2: (a) Data generation process: We create a dataset $\mathcal{D}$ consisting of $K$ item-label $(x_i, \ell_i)$ pairs where each $x_i \sim \mathcal{N}(0, 1/D)$ and $\ell_i$ is randomly sampled from $\{-1, +1\}$. The network receives a sequence of $N + 1$ tokens. Each of the first $N$ tokens is a concatenation $x_i \oplus \ell_i$ of an $(x_i, \ell_i)$ pair sampled uniformly from $\mathcal{D}$ (details in main-text). The final $N + 1$ target token consists of only the item $x_i$ as its label component is zero-ed out. The network is trained to correctly predict the label of the last item. Model architecture: The input is normalized using LayerNorm, then fed to our network, which consists of a one-layer attention network followed by a 3-layer MLP. (b) ICL performance demonstrates a sharp transition as a function of data diversity $K$. Further, at the transition threshold $K^* \approx 10^4$, we observe bimodality where the model either memorizes or generalizes. (c) IWL accuracy vs $K$. IWL and ICL accuracies follow opposite trends. (d) ICL accuracy curves show that ICL performance plateaus at the beginning of training but undergoes a rapid transition as ICL is acquired. (e) ICL is transient, i.e., ICL accuracy gradually decreases to chance levels when the parameters in the attention head are heavily regularized.

dimensional random vector with components drawn i.i.d from $\mathcal{N}(0, 1/D)$ and is randomly assigned one of two labels, $\ell_i \in \{-1, +1\}$.

The data is presented to the network as a sequence of $N + 1$ tokens, where each token $t_j$ is the item concatenated with its label, $t_j = x_j \oplus \ell_j$. The first $N$ tokens in the sequence are drawn uniformly from $\mathcal{D}$. The $N + 1$th token (the target token) has an empty label vector. The target token is chosen uniformly randomly from the $N$ items in the context, so that there is an exemplar always present in the context. Given an input sequence, the network is trained to predict the label of the target token using a binary cross-entropy loss. Since the total number of item-label pairs ($K$) is finite, the network can either memorize each item's label (IWL), or it can learn to use the exemplar within the context to predict the correct label (ICL).

To measure ICL, we construct a test dataset $\mathcal{D}_{\text{test}}$ consisting of novel item-label pairs (sampled like $\mathcal{D}$) and evaluate the network's accuracy on sequences sampled from $\mathcal{D}_{\text{test}}$ using the previously described procedure. To measure IWL, we evaluate the network on sequences sampled from $\mathcal{D}$, except that the target has no corresponding exemplar in the sequence. In this case, the context has no useful information, and the network must rely on the label's information encoded within its weights.

## 3 RESULTS

### 3.1 A ONE-LAYER TRANSFORMER MODEL RECAPITULATES ICL PHENOMENOLOGY

We begin with a one-layer attention-based network followed by a multi-layer perceptron (MLP). Given tokens $t_1, t_2, \ldots, t_{N+1}$, we first apply a LayerNorm operation to obtain $t_i' = \text{LayerNorm}(t_i)$.

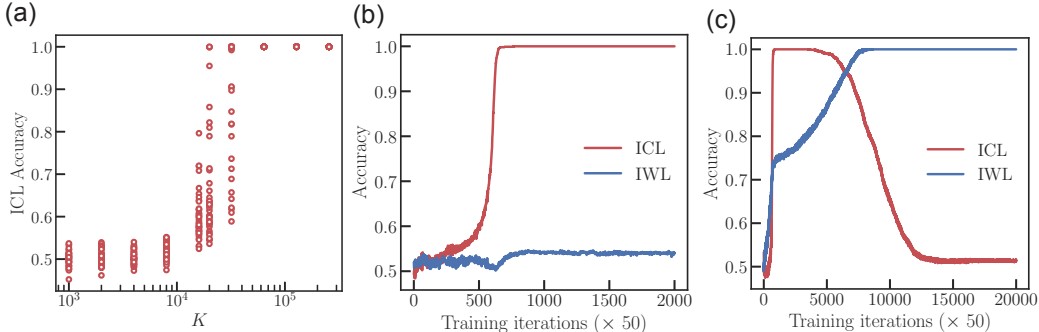

Figure 3: Phenomenology of the minimal model. (a) ICL performance in the minimal model demonstrates a sharp transition as a function of data diversity $K$. (b) ICL acquisition is abrupt during training. (c) ICL is transient when $w$ is exclusively regularized.

The attention operation then produces the output

$$u = t'_{N+1} + \sum_{j=1}^{N+1} \frac{e^{t_j^{'T} K^T Q t'_{N+1}}}{\sum_k e^{t_k^{'T} K^T Q t'_{N+1}}} V t'_j, \tag{1}$$

where $Q, K, V$ are $(D+2) \times (D+2)$ query, key and value matrices ($\ell_j$'s are one-hot vectors for the one-layer model, and we use $\ell_j = \pm 1$ elsewhere). The MLP $\phi$ takes input $u$ of dimension $D+2$ and produces logits $\phi(u)$. If the target's true label is $(1,0)$, the loss is $-\log \sigma(\phi(u))$, and if the target's true label is $(0,1)$, the loss is $-\log \sigma(-\phi(u))$, where $\sigma$ is the logistic function. The MLP is a three-layer ReLU network with hidden dimension $d$. We use stochastic gradient descent (SGD) with batch size 128, learning rate 0.01, weight-decay $10^{-10}$, $D = 63$, $N = 100$ and $d = 512$, unless otherwise specified.

We examine how ICL accuracy scales with the number of item-label pairs, $K$. For $K \ll K^*$, where $K^* \sim 10^4$, we find that the network shows IWL but fails to acquire ICL (Figure 2b,c). Conversely, for $K \gg K^*$, the network consistently acquires ICL but does not show IWL. The transition from IWL to ICL is sharp. Solutions in the vicinity of $K = K^*$ are bimodal (visible in Figure 2b and quantified later in Figure 6c). Examining the ICL accuracy during learning shows an abrupt transition from chance-level to perfect accuracy (Figure 2d). Previous work has also shown that ICL accuracy gradually decays to zero if training is continued for a sufficiently large number of iterations (Singh et al. (2023)). Transience is recapitulated when we significantly extend the number of training iterations (Figure 2e), provided the parameters within the attention head are regularized more heavily than those in the MLP. This skewed regularization scheme hints at a potential explanation for what causes transience (as noted in Singh et al. (2023)), which we will later explain quantitatively using our theoretical framework.

## 3.2 DISENTANGLING ICL AND IWL IN A MINIMAL MODEL

Figure 2(c-f) shows that, despite its simplicity, the one-layer transformer model captures the core features of the memorization to generalization transition and ICL training dynamics observed in more complex models. However, a mechanistic analysis is still challenging due to the nonlinearities in the attention head, the MLP and how these two operations interact. To make progress, we further reduce our one-layer transformer model into a disentangled model (which we refer to as the "minimal model" hereafter) by proposing two ansatz. We will show empirically that the minimal model also reproduces the phenomena in Figure 2(c-f). This minimal model is amenable to a theoretical analysis and leads to specific quantitative predictions. We then validate our ansatz by empirically testing these predictions using our original transformer model (Section 4).

To motivate the ansatz, we observe that ICL in this task involves a simple match-to-sample operation implemented by the attention head. The attention paid by the target token is determined by its dot-product similarity with the content (the first $D$ dimensions) of the tokens in the context. The value matrix reads the labels of the tokens weighted by the attention paid to those tokens and passes it on

to the MLP. Put together, these observations suggest that the relevant operations performed by the query-key product and the value matrix are captured by

$$K^T Q = \begin{pmatrix} \beta I_{D \times D} & 0_{D \times 1} \\ 0_{1 \times D} & 0_{1 \times 1} \end{pmatrix}, \quad V = \begin{pmatrix} 0_{D \times D} & 0 \\ 0 & w \end{pmatrix} \tag{2}$$

where $I_{D \times D}$ is a $D$-dimensional identity matrix, $\beta, w$ are (learnable) scalars and the rest of the components are zeroes.

From equation 1, the MLP receives the sum of the target token $t_{N+1}$ and the output of the attention head as its input. Note that all the information required by the MLP to memorize the label of the target is present in $t_{N+1}$. That is, IWL does not require the output of the attention head. Similarly, all the information required to predict the target's label using ICL is contained in the output of the attention head. Based on these observations, we posit that the final logit used to predict the target's label is the *sum* of logits generated independently by an MLP (which takes $x_{N+1}$ as input) and the attention head (using the simplified $K^T Q$ and $V$ matrices in equation 2). Specifically, given an input sequence $t_1, t_2, \ldots, t_{N+1}$, we assume the MLP $\phi$ produces a logit $z_{\mathrm{MLP}} = \phi(x_{N+1})$ and the attention head produces the logit

$$z_{\mathrm{ATT}} = \sum_{j=1}^{N} \frac{e^{\beta x_j^T x_{N+1}}}{\sum_{k=1}^{N} e^{\beta x_k^T x_{N+1}}} w \ell_j, \tag{3}$$

Put together, the predicted probability that the target's label is $+1$ is given by $\sigma(z_{\mathrm{MLP}} + z_{\mathrm{ATT}})$. The binary cross-entropy loss for a given input sequence is then $-\log \sigma(\ell_c(z_{\mathrm{MLP}} + z_{\mathrm{ATT}}))$, where $\ell_c = \pm 1$ is the true label of the target.

In summary, we assume an *independence* ansatz, where the attention head and MLP perform ICL and IWL respectively, and additively contribute towards the prediction of the target's label. Further, we assume that the majority of the ICL learning dynamics is captured by reducing the $K^T Q$ and $V$ matrices to two "order parameters", $\beta$ and $w$. That the *strengths* of the relevant attention operations determine ICL acquisition is our second ansatz.

The minimal model is parameterized by $\beta, w$ and the parameters of the MLP (a three-layer ReLU network of hidden dimension 512). The model is trained and evaluated using the same procedures used on the transformer model. Figure 3(a-c) show that the three phenomena of interest, that is, the transition from memorization to generalization, abrupt ICL learning and ICL transience, are reproduced by the minimal model. We now use the minimal model to develop an analytical theory. We outline the main results here and present more detailed derivations in the Appendix.

### 3.3 The loss landscape of the minimal model

We consider the asymptotic limit $K \gg N \gg 1$ and the infinite-dimensional limit $D \to \infty$ (recall, $K > 10^3, N = 10^2, D = 63$ in our experiments). From equation 3, ICL is acquired when $w, \beta \gg 1$. Our goal is to compute the time taken for the network to acquire ICL starting from $w = w_0, \beta = \beta_0$ with $|w_0|, |\beta_0| \ll 1$.

In the limit $D \to \infty$, the dot product $x_j^T x_{N+1}$ is 1 if $x_j$ is a copy of the target and 0 otherwise. It is unlikely there is more than one copy of the target in the context when $K \gg N$. Let $c$ denote the index of this copy. From equation 3, we have

$$z_{\mathrm{ATT}} \approx w \left( \frac{e^{\beta}}{e^{\beta} + N - 1} \ell_c + \frac{1}{e^{\beta} + N - 1} \left( 2n_+ - N - \ell_c \right) \right), \tag{4}$$

where $n_+$ is the (binomally distributed) number of tokens with label $+1$ amongst the $N$ tokens in the context. When $N \gg 1$, $n_+/N \approx 1/2 + \eta/2\sqrt{N}$, where $\eta \sim \mathcal{N}(0, 1)$.

Next, the MLP's contribution to the average loss appears only through the *distribution* of logits obtained by applying the MLP to each of the $K$ items in $\mathcal{D}$. In particular, denote $P^+$ as the distribution of logits obtained when the MLP is applied to the items in $\mathcal{D}$ with a $+1$ label. We use the fact that the two labels are symmetric, and average over $n_+$ and $P^+$ to show that the average binary cross-entropy loss $\mathcal{L}$ is (see Appendix)

$$\mathcal{L} \approx -\left\langle \left( 1 + \frac{\eta}{\sqrt{N}} \right) \log \sigma \left( \phi^+ + w \left( \frac{e^{\beta} - 1}{e^{\beta} + N - 1} + \frac{\eta \sqrt{N}}{e^{\beta} + N - 1} \right) \right) \right\rangle_{\eta \sim \mathcal{N}(0,1), \phi^+ \sim P^+} . \tag{5}$$

The loss landscape throughout training is thus specified by $P^+$, $w$ and $\beta$. Retaining the fluctuations in $\eta$ is necessary to accurately describe ICL acquisition.

### 3.4 THE DYNAMICS OF ICL ACQUISITION

To examine the dynamics of ICL acquisition, we find an expression for the loss when $|w| \ll \sqrt{N}$ and $e^\beta - 1 \ll N$ (for arbitrary $P^+$). Both these conditions are satisfied at initialization ($|w_0|, |\beta_0| \ll 1$). From equation 5, a few steps of simplification leads to (Appendix)

$$\mathcal{L} \approx \left\langle \log(1 + e^{-\phi^+}) \right\rangle_{\phi^+} - \frac{c_1}{N} \left( e^\beta w - \frac{c_2 w^2}{2} \right), \tag{6}$$

where $c_1 \equiv \langle \sigma(-\phi^+) \rangle_{\phi^+}$ and $c_2 \equiv 1 - \langle \sigma(-\phi^+)^2 \rangle_{\phi^+}/c_1$. Here, we used $|w| \ll \sqrt{N}$ and $e^\beta - 1 \ll N$ to Taylor expand equation 5 and retained terms to order $1/N$ (terms of order $1/\sqrt{N}$ vanish in expectation). The distribution from which $\phi^+$ is drawn has been dropped for notational convenience.

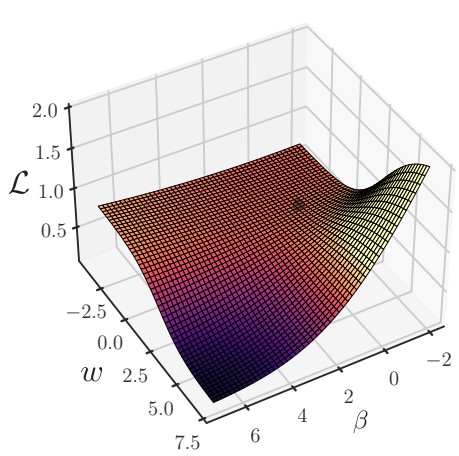

Equation 6 allows us to make several important inferences. The first term in the r.h.s of equation 6 is the loss incurred by the MLP. It does not involve $w, \beta$ and thus does not affect ICL learning dynamics. Since the second term in the r.h.s of equation 6 is small at initialization, the rate at which the MLP memorizes is not affected by ICL learning. That is, IWL proceeds without any competition from ICL until ICL is acquired (which happens abruptly).

The scalar variables $c_1$ and $c_2$ depend on $P^+$ and thus depend on the time $t$ since training began. Their evolution in general depends on multiple factors, including MLP architecture, initialization scheme and the number of tasks $K$ to be memorized. Importantly, IWL influences ICL acquisition only through $c_1(t)$ and $c_2(t)$, which in turn depend only on how the MLP memorizes class labels. We proceed with our analysis by retaining $c_1(t)$ and $c_2(t)$ as yet-to-be-determined MLP-specific dynamical "order parameters", keeping in mind that their dependence on $t$ and $K$ will play an important role in our analysis further below.

Figure 4: The approximate ICL loss landscape $\mathcal{L}$ (fixing $z_{\text{MLP}} = 0$) in the minimal model as a function of the key parameters $\beta, w$ exhibits a nearly flat region close to initialization, but the dynamics always leads to ICL acquisition ($w, \beta \gg 1$).

Gradient descent dynamics over the loss in equation 6 gives

$$\frac{dw}{dt} = \frac{c_1}{N} \left( e^\beta - c_2 w \right), \tag{7}$$

$$\frac{d\beta}{dt} = \frac{c_1}{N} \left( w e^\beta \right). \tag{8}$$

Learning initially proceeds at a slow rate $c_1/N$ (since $N \gg 1$ and $0 < c_1 < 1$). Since $\phi^+$ on average increases as the MLP memorizes, $c_1$ decreases and slows down ICL acquisition. If the MLP (near) perfectly memorizes the $K$ item-label pairs before ICL is acquired, then ICL is never acquired. In other words, the loss "explained away" due to MLP memorization creates an effective competition between IWL and ICL acquisition despite the additive contributions of the MLP and the attention head to the logit. Since $0 < c_2 < 1$, equation 7 shows that $w$ eventually converges from its initial value to a positive value $w = e^\beta/c_2$. $\beta$ increases monotonically when $w$ is positive until ICL is acquired. Thus, equation 7 and equation 8 imply that ICL will always be acquired, however slowly, if the MLP is unable to perfectly memorize the $K$ item-label pairs (i.e., $c_1(\infty) > 0$).

However, the choice of label statistics in the context matters. For example, consider the case when $N$ is even and there are exactly $N/2$ tokens with $+1$ and $-1$ labels in the context. To compute the mean loss $\mathcal{L}'$ in this scenario, we set $\eta = 0$ in equation 5 and Taylor expand w.r.t $(e^\beta - 1)/N$ to get

$$\mathcal{L}' \approx \left\langle \log(1 + e^{-\phi^+}) \right\rangle_{\phi^+} - \frac{c_1}{N} w \left( e^\beta - 1 \right). \tag{9}$$

The origin is a saddle point. The parameters either flow to the ICL solution ($w, \beta > 0$) or to a suboptimal solution ($w, \beta < 0$) depending on the initial values $w_0, \beta_0$. That is, ICL acquisition is not guaranteed. Intuitively, when $n_+$ is binomially distributed, the network learns that the target's label is more likely to be a $+1$ if there are more $+1$ labels than $-1$ labels in the context. This bias, however small, pushes $w$ to a positive value and leads the network into the ICL basin. When the numbers of $+1$ and $-1$ labels are forced to be equal, $w$ could flow into the basin that leads to ICL or flow to an alternative potentially suboptimal solution. We revisit the $n_+ = N/2$ case in Section 4.

### 3.5 Exponential dependence of $t_{\text{ICL}}$ on initial conditions

Equation 7 and equation 8 allow us to estimate the number of iterations it takes to acquire ICL (Appendix). Note that "time" $t$ here is a proxy for the number of iterations, which we can only determine up to a constant pre-factor. Exact integration of equations 7 and 8 is infeasible, but an approximate expression is obtained when $|w_0|, |\beta_0| \ll 1$. We fix $w_0 = 0$ hereafter, though the more general case of $w_0 \neq 0$ can be solved (Appendix). We find that the number of iterations it takes for ICL acquisition (denoted $\tau_K$) satisfies

$$N\sqrt{2\pi}e^{-\beta_0} \approx I_K(\tau_K), \quad \text{where } I_K(t) \equiv 2\int_0^t c_1(t')dt'. \tag{10}$$

The subscript $K$ is introduced to highlight that $c_1$ depends on $K$.

We first consider the case $K = \infty$ so that the MLP is unable to memorize $\mathcal{D}$. The MLP logit $\phi_+$ is distributed symmetrically around 0, in which case $c_1(t) = \langle \sigma(-\phi^+) \rangle_{\phi^+} \approx 1/2$ and $I_\infty(t) = t$. Solving for $\tau_\infty$ (which we call $t_{\text{ICL}}$ hereafter) using equation 10, we get

$$t_{\text{ICL}} \approx N\sqrt{2\pi}e^{-\beta_0}. \tag{11}$$

The dynamics are qualitatively different when $-\beta_0$ is large and $e^{\beta_0} \ll 1$. In this case, we obtain $t_{\text{ICL}} \approx Ne^{-2\beta_0}$ (Appendix). We numerically verify the exponential dependence of $t_{\text{ICL}}$ on the initial values of $\beta_0$ (Supplementary Figure A.1a). A consequence of this exponential dependence is that normal-distributed values of $\beta_0$ will lead to a long-tailed distribution of $t_{\text{ICL}}$. In pictorial terms, due to the nearly flat loss landscape close to initialization (Figure 4), small variation in the initial parameters $w_0, \beta_0$ leads to large variation in when ICL is acquired.

### 3.6 Memorization scaling laws and the transition from memorization to generalization

Equation 10 shows that the behavior of an MLP-specific quantity, $c_1(t)$ (via $I_K$), determines when ICL is acquired for different values of $K$. It is useful to introduce the quantity $I_K(\infty)$, which can be interpreted as the time taken for the MLP to memorize a dataset of size $K$. Equations 10 and 11 together with the monotonicity of $I_K(t)$ imply that ICL is acquired if

$$t_{\text{ICL}} < I_K(\infty). \tag{12}$$

We delineate two distinct mechanisms depending on whether $I_K(\infty)$ is finite or not:

1. Capacity-constrained: We call the network capacity-constrained if $I_K(t)$ diverges as $t \to \infty$, i.e., the network never fully memorizes the dataset. Equation 12 then implies that the network generalizes when $K > K_{\text{cc}}$, where $K_{\text{cc}}$ is the smallest $K$ at which the network is capacity-constrained.

2. Differential learning kinetics: It is possible that $I_K(\infty)$ is finite. In this case, the network transitions from memorization to generalization at $K = K^*$ such that $t_{\text{ICL}} \approx I_{K^*}(\infty)$. In other words, when $K > K^*$, it takes longer for the network to memorize the dataset (even though it has the capacity to do so) than it takes for the network to generalize. We call this case the *differential learning kinetics* regime as the relative *rates* at which the network memorizes and generalizes determine when the transition occurs.

The divergence of $I_K(t)$ as $t \to \infty$ may occur either because the network has limited capacity to memorize the $K$ samples or because of the data distribution. For example, if the rank-frequency distribution of item-label pairs follows a Zipf's law $p(f) \sim f^{-\alpha}$ with exponent $\alpha \leq 1$, then the network's loss is dominated by rare item-label pairs that are not memorized. Previous work has shown that such skewed data distributions indeed favor ICL acquisition (Chan et al. (2022)).

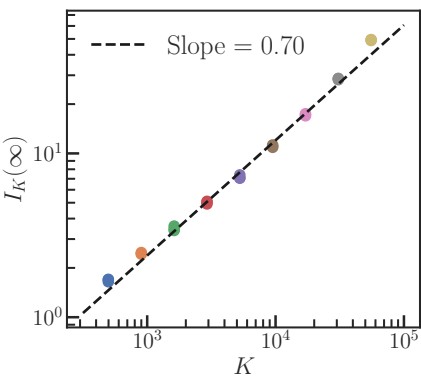

Figure 5: $I_K(\infty)$ shows a power-law scaling with $K$ with exponent $\nu \approx 0.7$.

We examine the behavior of $I_K$ for a uniform distribution over item-label pairs. To our knowledge, current deep learning theory does not inform how the distribution of logits (or summary statistics such as $c_1$) scales with $t$ and $K$ for typical MLP architectures. To make progress, we empirically measure $c_1(t), c_2(t)$ for different $K$ using an independent set of MLP experiments (Supplementary Figure A.2). We find that $c_1(t)$ for $K$ between 500 and 50000 decays fast enough such that its integral $I_K(\infty)$ is finite (Supplementary Figure A.3a). Our MLP is thus not capacity-constrained. Further, we uncover a scaling law, $I_K(\infty) \sim K^\nu$, where $\nu \approx 0.7$ (Figure 5). From equation 12, using the expression for $t_{\text{ICL}}$ in equation 11 and the scaling law $I_K(\infty) \sim K^\nu$ leads to an estimate for the task diversity threshold $K^*$ for the transition from memorization to generalization:

$$K^* \sim N^{1/\nu} e^{-\beta_0/\nu}, \tag{13}$$

up to a constant prefactor. Note the exponential dependence on $\beta_0$, whose random initialization implies that there is a range of $K$ for which the network will either show memorization or generalization. The theory further predicts a non-trivial power-law relation between the task diversity threshold and the context length.

### 3.7 Slow IWL explains transient ICL

We now explain why transience appears in our minimal model (Figure 3c) when the attention head is regularized more heavily compared to the MLP. For simplicity, we impose $L_2$ regularization with parameter $\lambda_w$ only on $w$. We return to equation 5 for the loss, which applies throughout training. Since $w, \beta \gg 1$ after ICL is acquired, we can simplify equation 5 to obtain (Appendix)

$$\mathcal{L} \approx \left\langle e^{-\phi^+} \right\rangle_{\phi^+} e^{-w} + \frac{\lambda_w w^2}{2}. \tag{14}$$

Once ICL is acquired, memorization slows down dramatically due to the small factor $e^{-w}$. Without $L_2$ regularization on $w$, $w$ continues to increase (at a decreasing rate) and ICL is not transient. However, when $w$ is regularized, $w$ after ICL acquisition tracks $w_{\text{tr}}$, where

$$w_{\text{tr}}(t) \approx W\left(c_3(t)/\lambda_w\right), \quad c_3(t) \equiv \left\langle e^{-\phi^+} \right\rangle_{\phi^+}. \tag{15}$$

The Lambert W function $W(x)$ is monotonic in $x$ when $x$ is positive. $c_3$ decreases as the network memorizes the dataset (Figure A.2c). Thus, $w_{\text{tr}}$ decreases as $c_3$ decreases. $w_{\text{tr}}$ decays to zero (and ICL fades away) when the dataset is sufficiently memorized, i.e., when $c_3 \ll \lambda_w$. Thus, the analysis suggests that extremely slow memorization coupled with regularization leads to ICL transience. We note however that in more complex models the effects of a global regularization parameter on different sub-circuits are hard to disentangle, which may explain the puzzling observations in Singh et al. (2023).

Equation 15 hints at a relationship between the loss on ICL sequences ($\mathcal{L}_{\text{ICL}}$) and the loss of IWL sequences ($\mathcal{L}_{\text{IWL}}$) after ICL is acquired. We use a heuristic argument (Appendix) to show that

$$\mathcal{L}_{\text{IWL}} \approx -\frac{1}{2}\log(\mathcal{L}_{\text{ICL}}), \quad \text{when } \mathcal{L}_{\text{ICL}} \ll 1,$$

$$\mathcal{L}_{\text{ICL}} \approx -\frac{1}{2}\log(\mathcal{L}_{\text{IWL}}), \quad \text{when } \mathcal{L}_{\text{IWL}} \ll 1. \tag{16}$$

These approximate relations between $\mathcal{L}_{\text{ICL}}$ and $\mathcal{L}_{\text{IWL}}$ are consequences of our two ansatz. If our ansatz are valid, the theory predicts that these relations should hold from the moment ICL is acquired until it fades due to gradual IWL.

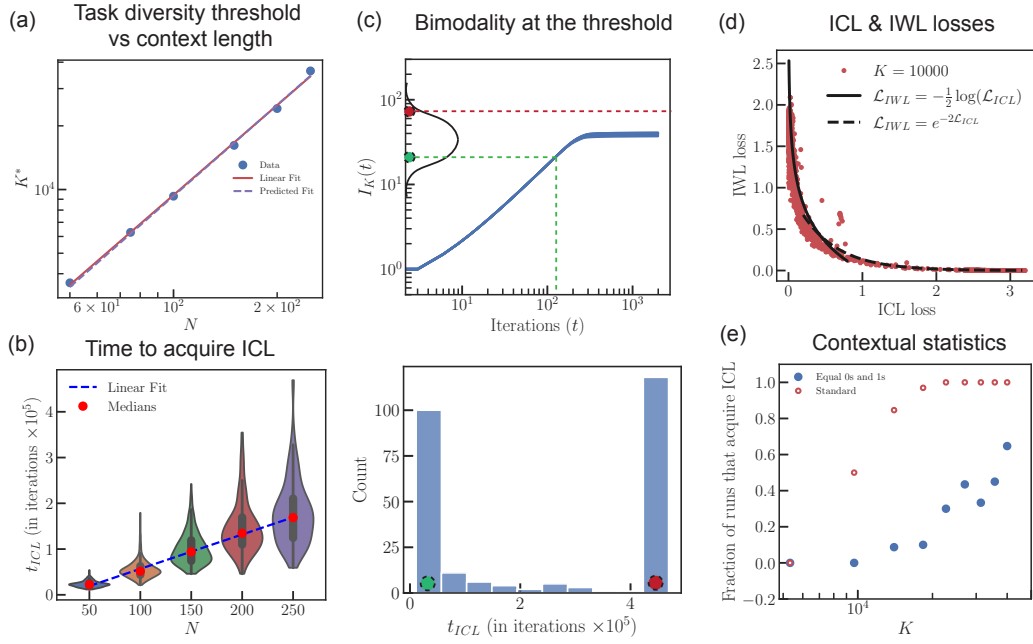

Figure 6: (a) The critical task diversity threshold $K^*$ exhibits a power law relationship with respect to $N$. The experimentally determined critical exponent (linear fit) closely matches our predicted critical exponent $1/\nu$ where $\nu \approx 0.7$ (predicted fit). (b) The median time taken to acquire ICL ($t_{\text{ICL}}$) scales linearly as a function of $N$ and the distribution of $t_{\text{ICL}}$ is long-tailed. For clearer visualization of the full distribution, see Supplementary Figure A.4. (c) Close to $K = K^*$, depending on the initialization seed (red and green dots), the network will either generalize (green) or not (red). Different seeds are predicted to show either small $t_{\text{ICL}}$ or do not acquire ICL, with few intermediates. A histogram of $t_{\text{ICL}}$ for $K = 9666$ confirms this prediction at the critical task diversity threshold $K^*$. Note that the maximum number of iterations is $\approx 4.5 \times 10^5$. (d) In runs that exhibit transience, we observe a relationship between ICL loss and IWL loss after the model has acquired ICL, closely matching our predicted functional relationship. (e) We measured the fraction of solutions that acquired ICL as we vary $K$, by training at least 20 models with different seeds for each $K$, on a new dataset $\mathcal{D}'$ where every sequence is constrained to contain exactly $N/2$ items of each label. We observe that the critical task diversity threshold $K^*$ is greatly increased and that many more solutions fail to achieve ICL compared to models trained on our standard dataset $\mathcal{D}$.

## 4  EMPIRICAL VALIDATION

The theory makes a number of new quantitative predictions related to ICL acquisition. We empirically test six nontrivial predictions that span various aspects of ICL phenomenology using the original transformer model in equation 1.

**Power-law scaling of the task diversity threshold with context length.** Equation 13 predicts a highly non-trivial power law relationship between the task diversity threshold $K^*$ and $N$. To test this prediction, we train our original transformer model (equation 1) at varying $N$ and $K$. At each $N$, we observe a sharp transition from memorization to generalization as $K$ increases (Supplementary Figure A.7). For each $N$, we determine $K^*$ by fitting a sigmoidal curve to ICL performance as a function of $K$. As $\nu \approx 0.7$, Equation 13 predicts an exponent of $1/\nu \approx 1.43$, closely matching our measured exponent $\approx 1.41$ (Figure 6a).

**Linear scaling of the time taken to acquire ICL with context length.** Equation 11 predicts that $t_{\text{ICL}}$ (time taken to acquire ICL) scales linearly with $N$. To test this, we train our original transformer model (equation 1) at varying $N$ and take the limit $K \to \infty$ by resampling our dataset $\mathcal{D}$ at every training iteration. We then determine $t_{\text{ICL}}$ as the epoch at which ICL accuracy exceeds 95%. We train $\approx 100$ seeds for each $N$ to obtain the full distribution of $t_{\text{ICL}}$. Figure 6b confirms a linear

relationship between the median $t_{\text{ICL}}$ as a function of $N$. We verified that the linear relationship also holds for a two-layer transformer (Supplementary Figure A.5).

**Bimodal solutions near the transition.** For $K$ near the transition, $K \approx K^*$, equation 13 predicts that independent runs will either have $t_{\text{ICL}} > I_{K^*}(\infty)$ or $t_{\text{ICL}} < I_{K^*}(\infty)$ depending on the network's initial parameters. In the former case, ICL is never acquired. The nonlinear form of $I_K(t)$ (illustrated in Figure 6c, top) suggests that in the latter case, $t_{\text{ICL}}$ will be short, with few intermediates in between. We verify this bimodal behavior of the solutions around the transition by generating runs with $\approx 250$ seeds near $K \approx K^*$ (Figure 6c, bottom).

**Long-tailed distribution of the time taken to acquire ICL.** Equation 11 predicts a long-tailed distribution of $t_{\text{ICL}}$. We verify this prediction in histograms of $t_{\text{ICL}}$ for each $N$, using the same data as in Figure 6b (Supplementary Figure A.4).

**Interdependence of the ICL loss and IWL loss after ICL acquisition.** Equation 16 predicts a non-trivial relationship between the ICL loss $\mathcal{L}_{\text{ICL}}$ and the IWL loss $\mathcal{L}_{\text{IWL}}$ after ICL is acquired. In runs in which we observed transience, we plot IWL performance as a function of ICL performance and observe a close match between equation 16 and data (Figure 6d).

**Failure of ICL acquisition when every sequence contains $N/2$ tokens of each label.** Equation 9 predicts that the network is more likely to fail to acquire ICL when there are exactly $N/2$ tokens with $+1$ and $-1$ labels in the context. We test this prediction in a dataset $\mathcal{D}'$ with $N = 100$, where each training sequence (including the target token) viewed by the one-layer transformer has $N/2$ tokens of each label in its context. For each $K$, we trained the model using $\approx 20$ seeds. We measure the probability of acquiring ICL as the fraction of seeds whose final ICL performance exceeds 75%. We observe much lower probability of acquiring ICL for models trained on $\mathcal{D}'$ compared to those trained on our standard $\mathcal{D}$ (Figure 6e). Moreover, the loss curves across seeds are diverse, with many seeds not acquiring ICL and some saturating at sub-optimal solutions (See Supplementary Figure A.8).

## 5 CONCLUSION

Here, we propose a theory based on the ansatz that the network contains sub-circuits that are independently involved in memorization and generalization. A trade-off arises simply because the rate at which one sub-circuit is optimized depends on how much loss is already explained by the other sub-circuit(s). Building on this theory, we show that the transition from memorization to generalization in our model is determined by the relative rates at which these sub-circuits memorize and generalize. However, the theory does not rule out the possibility that capacity constraints play a role in other scenarios.

This ansatz, despite being cast in the context of a simplified one-layer model, explains a surprising variety of ICL-related phenomena observed with much larger models. These include a long-tailed distribution in when ICL is acquired, an MLP memorization scaling law, the bimodality of solutions at the task diversity threshold, the transient nature of ICL, amongst other novel quantitative relations that our theory identifies. The two most striking predictions are (1) the non-trivial relationship between an MLP-specific memorization scaling law ($I_K(\infty) \sim K^\nu$) and a task diversity threshold scaling law w.r.t context length ($K^* \sim N^{1/\nu}$), and (2) the long-tailed distribution of when ICL is acquired and its linear scaling with context length ($t_{\text{ICL}} \sim N$). Both these predictions have been validated in our experiments.

Our results offer some hope that seemingly intractable phenomena observed in large models can be reproduced and analyzed using simpler, tractable models through careful experimental design. However, further work is necessary to examine to what extent such insights provided by small models remain valid for larger models (that potentially contain many sub-circuits) and for more naturalistic tasks (where a clear distinction between memorization and generalization cannot be made) (Min et al. (2022); Wei et al. (2023); Pan (2023); Shi et al. (2024)).

## ACKNOWLEDGMENTS

We thank Kenneth A. Norman, Albert Qin, Cole Gibson and anonymous reviewers for support and valuable feedback on the manuscript. AN is supported by NIH grant RF1MH125318. GR is partially supported by a joint research agreement between Princeton University and NTT Research Inc.

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

## A  APPENDIX

We present a detailed analysis of the minimal model outlined in the main text. Suppose the $(D+1)$-dimensional input tokens are $t_1, t_2, \ldots, t_{N+1}$, where $t_i = (x_i, \ell_i)$ for $i \leq N$ and $t_{N+1} = (x_{N+1}, 0)$. In the minimal model, we consider two logits $z_{\text{MLP}} = \phi(x_{N+1})$ and

$$z_{\text{ATT}} = \sum_{j=1}^{N} \frac{e^{\beta x_j^T x_{N+1}}}{\sum_{k=1}^{N} e^{\beta x_k^T x_{N+1}}} w \ell_j. \tag{17}$$

$\phi$ is a multi-layer perceptron (MLP), which throughout our paper is a three-layer ReLU network with hidden dimension 512. The final logit $z$ used for classification is the sum of the contributions from the MLP and the attention head, $z = z_{\text{MLP}} + z_{\text{ATT}}$. The minimal model can be obtained from the one-layer transformer model by (1) removing the LayerNorm operation, (2) the interaction strength ansatz, i.e., by assuming the query, key and value matrices in the attention head have the form

$$K^T Q = \begin{pmatrix} \beta I_{D \times D} & 0_{D \times 1} \\ 0_{1 \times D} & 0_{1 \times 1} \end{pmatrix}, \quad V = \begin{pmatrix} 0_{D \times D} & 0 \\ 0 & w \end{pmatrix}, \tag{18}$$

and (3) the independence ansatz, where the residual term $x_{N+1}$ is processed by the MLP to produce $z_{\text{MLP}}$, the output of the attention head is $z_{\text{ATT}}$ and these two logits are summed to produce the final logit $z$. However, we stress that the minimal model serves as a phenomenological model and is not derived from the one-layer transformer model.

To reproduce the phenomenology shown in Figure 3, we optimize $\beta, w$ and the parameters of the MLP $\phi$ using the same procedure used to train the full model. In particular, we use stochastic gradient descent (SGD) with batch size 128, learning rate 0.01, weight-decay $10^{-10}$, $D = 63$, $N = 100$ and MLP hidden dimension $d = 512$. To reproduce transience in Figure 3c with a fewer number of iterations, we increase the weight-decay parameter on $w$ and $\beta$ to $10^{-3}$.

### A.1  THEORETICAL ANALYSIS OF THE MINIMAL MODEL

To derive an analytical expression for the loss landscape of the minimal model, we assume (1) that only one copy (say $x_c$) of the target $x_{N+1}$ is present in the context, (2) that $x_{N+1}.x_i = 1$ if $i = c$ and 0 otherwise, and (3) that the distributions of logits obtained when the MLP is applied to all the

items in $\mathcal{D}$ with labels $+1$ and $-1$, (say, $P^+(\phi)$ and $P^-(\phi)$ respectively) are identical except for the sign: $P^+(\phi) = P^-(-\phi)$.

These three assumptions are justified when $K \gg N \gg 1$ and $D \to \infty$. Note that in the simulations presented in Figure 3, we use $K > 10^3$, $N = 100$ and $D = 63$. Assumption (1) is justified when $K \gg N$ as it is unlikely that more than one copy of the target is sampled from the dataset $\mathcal{D}$ ($|\mathcal{D}| = K$) in a single context of length $N$. Assumption (3) is justified when $K \gg 1$ as all the items in $\mathcal{D}$ are statistically identical irrespective of the label. Assumption (2) is justified when $D \to \infty$ based on our sampling process, which will ensure that in this limit the norm of each item is one and different items are orthogonal. For finite $D$, our theory will break down when $N$ is sufficiently large, though the precise scaling relationship between $D$ and $N$ for when the theory will break down remains to be examined.

Let $c$ be the index of the target's copy in the context. Under the assumptions stated above,

$$
z_{\text{ATT}} = w \left( \frac{e^\beta}{e^\beta + N - 1} \ell_c + \frac{1}{e^\beta + N - 1} \sum_{j \neq c} \ell_j \right)
$$
$$
= w \left( \frac{e^\beta}{e^\beta + N - 1} \ell_c + \frac{1}{e^\beta + N - 1} \left( 2n_+ - N - \ell_c \right) \right), \tag{19}
$$

where $n_+$ is the number of tokens which have label $+1$ amongst the $N$ tokens in the context. Suppose we split the dataset $\mathcal{D}$ into two datasets $\mathcal{D}^+$ and $\mathcal{D}^-$ containing items with $+1$ and $-1$ labels respectively. The MLP (at any particular stage of training) when applied to the items from $\mathcal{D}^\pm$ produces two histograms of logits $P^\pm(\phi)$.

Recall that the binary cross-entropy loss for a given input sequence is $-\log \sigma(\ell_c(z_{\text{MLP}} + z_{\text{ATT}}))$. The average loss is obtained by averaging over the cases when the target is drawn from $\mathcal{D}^+$ and $\mathcal{D}^-$, and by taking an expectation over $n_+$ (which follows a binomial distribution, $B(n_+)$). Given that there are $n_+$ items with label $+1$ in the sequence, the target's label is $\ell_c = \pm 1$ with probability $n_+/N$ and $1 - n_+/N$ respectively. The average binary cross-entropy loss is then

$$
\mathcal{L} = -\sum_{n_+=0}^{N} \frac{n_+}{N} B(n_+) \left\langle \log \sigma \left( \phi^+ + w \left( \frac{e^\beta}{e^\beta + N - 1} + \frac{1}{e^\beta + N - 1} \left( 2n_+ - N - 1 \right) \right) \right) \right\rangle_{\phi^+ \sim P^+}
$$
$$
- \sum_{n_+=0}^{N} \frac{N - n_+}{N} B(n_+) \left\langle \log \sigma \left( -\phi^- + w \left( \frac{e^\beta}{e^\beta + N - 1} + \frac{1}{e^\beta + N - 1} \left( N - 2n_+ - 1 \right) \right) \right) \right\rangle_{\phi^- \sim P^-},
\tag{20}
$$

where the two sums consider the cases when the target's label is $+1$ and $-1$. Given the symmetry between positive and negative labels, the contribution to the loss when $\ell_c = -1$ (the second sum) is approximately equal to the loss when $\ell_c = +1$ (the first sum). Formally, using $P^+(\phi) = P^-(-\phi)$, replacing $n_+$ with $N - n_+$ in the second sum and using the property $B(n_+) = B(N - n_+)$, we get

$$
\mathcal{L} \approx -2 \sum_{n_+=0}^{N} \frac{n_+}{N} B(n_+) \left\langle \log \sigma \left( \phi^+ + w \left( \frac{e^\beta}{e^\beta + N - 1} + \frac{1}{e^\beta + N - 1} \left( 2n_+ - N - 1 \right) \right) \right) \right\rangle_{\phi^+ \sim P^+}.
\tag{21}
$$

When $N \gg 1$, $n_+/N$ is approximately Gaussian-distributed: $n_+/N \approx 1/2 + \eta/2\sqrt{N}$, where $\eta \sim \mathcal{N}(0,1)$. This gives

$$
\mathcal{L} \approx -\left\langle \left( 1 + \frac{\eta}{\sqrt{N}} \right) \log \sigma \left( \phi^+ + w \left( \frac{e^\beta - 1}{e^\beta + N - 1} + \frac{\eta \sqrt{N}}{e^\beta + N - 1} \right) \right) \right\rangle_{\eta \sim \mathcal{N}(0,1), \phi^+ \sim P^+},
\tag{22}
$$

which is presented in the main text (equation 5).

## A.2 Expanding the loss before ICL acquisition

$w, \beta$ are initialized at $w_0, \beta_0$ where $|w_0|, |\beta_0| \ll 1$. Define $\gamma \equiv e^\beta - 1$. Close to initialization, $w, \beta$ satisfy $\gamma/N \ll 1$ and $|w|/\sqrt{N} \ll 1$. We compute the time it takes for the network to acquire ICL.

ICL is acquired when $w, \beta \gg 1$. From equation 22, since $\gamma \ll N$, we have

$$\mathcal{L} \approx - \left\langle \left(1 + \frac{\eta}{\sqrt{N}}\right) \log \sigma \left(\phi^+ + w \left(\frac{\gamma}{N} + \frac{\eta}{\sqrt{N}}\right)\right)\right\rangle_{\eta, \phi^+}, \tag{23}$$

where the distributions from which $\eta$ and $\phi^+$ are drawn has been dropped for notational convenience. Since $\gamma/N \ll 1$ and $|w|/\sqrt{N} \ll 1$, we have $w\left(\frac{\gamma}{N} + \frac{\eta}{\sqrt{N}}\right) \ll 1$. Using $\sigma(x) = 1/(1 + e^{-x})$ and two Taylor expansions, we get

$$\mathcal{L} \approx \left\langle \left(1 + \frac{\eta}{\sqrt{N}}\right) \log \left(1 + e^{-\phi^+ - w\left(\frac{\gamma}{N} + \frac{\eta}{\sqrt{N}}\right)}\right)\right\rangle_{\eta, \phi^+}, \tag{24}$$

$$\approx \left\langle \left(1 + \frac{\eta}{\sqrt{N}}\right) \log \left(1 + e^{-\phi^+} \left(1 - w\left(\frac{\gamma}{N} + \frac{\eta}{\sqrt{N}}\right) - \frac{w^2}{2}\left(\frac{\gamma}{N} + \frac{\eta}{\sqrt{N}}\right)^2\right)\right)\right\rangle_{\eta, \phi^+}, \tag{25}$$

$$\approx \left\langle \log(1 + e^{-\phi^+}) \right\rangle_{\phi^+} + \left\langle \left(1 + \frac{\eta}{\sqrt{N}}\right) \log \left(1 - \sigma(-\phi^+)\left(w\left(\frac{\gamma}{N} + \frac{\eta}{\sqrt{N}}\right) + \frac{w^2}{2}\left(\frac{\gamma}{N} + \frac{\eta}{\sqrt{N}}\right)^2\right)\right)\right\rangle_{\eta, \phi^+}, \tag{26}$$

$$\approx \left\langle \log(1 + e^{-\phi^+}) \right\rangle_{\phi^+} - \frac{c_1}{N}\left((1 + \gamma)w - \frac{c_2 w^2}{2}\right), \tag{27}$$

$$= \left\langle \log(1 + e^{-\phi^+}) \right\rangle_{\phi^+} - \frac{c_1}{N}\left(e^\beta w - \frac{c_2 w^2}{2}\right) \tag{28}$$

where we define the two MLP-specific dynamical variables,

$$c_1 \equiv \langle \sigma(-\phi^+) \rangle_{\phi^+}, \tag{29}$$

$$c_2 \equiv 1 - \langle \sigma(-\phi^+)^2 \rangle_{\phi^+} / c_1. \tag{30}$$

In the second to last step, we have expanded $\log(1 + x)$ for small $x$, computed the expectation over $\eta$ and ignored higher order terms as the $1/N$ term is the dominant contribution (the $1/\sqrt{N}$ contribution vanishes in expectation). Equation 28 is presented in the main text (equation 6).

Since $\sigma(-x) = 1 - \sigma(x)$, $c_1$ is one minus the probability of a correct classification, averaged over the dataset. At initialization, the logits are centered around $\phi^+ \approx 0$ so that $\sigma(\phi^+) \approx 1/2$ and therefore $c_1, c_2 \approx 1/2$. $\phi^+$ increases on average during training, which means $c_1$ decreases and $c_2$ increases during training. $c_1 = 0$ and $c_2 = 1$ in the idealized scenario where the dataset is perfectly memorized. We train an MLP in independent experiments to memorize $\mathcal{D}$ with the same hyperparameters used for training the one-layer transformer. We verify empirically that our MLP can near-perfectly memorize the dataset for dataset size $K$ up to $10^4$ (Figure A.2 and Supplementary Figure A.3a).

## A.3 ICL ACQUISITION

We consider ICL to be acquired when $w, \beta \gg 1$. To compute the time taken to acquire ICL, we consider gradient descent dynamics of $w$ and $\beta$ close to initialization. From equation 28, we get

$$\frac{dw}{dt} = \frac{c_1}{N}\left(e^\beta - c_2 w\right), \tag{31}$$

$$\frac{d\beta}{dt} = \frac{c_1}{N}\left(we^\beta\right). \tag{32}$$

Numerical simulations (Figure A.1) with $c_1, c_2$ fixed at $1/2$ reveal qualitatively different dynamics depending on whether $|\beta_0| \ll 1$ or $-\beta_0 \gg 1$. Note that $e^{\beta_0} - 1 \ll N$ in both these cases, so that our approximation equation 28 is still valid. We first examine the relevant case $|\beta_0| \ll 1$. The analysis of the second case $-\beta_0 \gg 1$ is presented further below for completeness.

Equations 31 and 32 cannot be solved exactly and we resort to approximations. When $|\beta_0| \ll 1$ and $|w_0| \ll 1$, we have $e^\beta \approx 1$ and $|w| \ll 1$ close to initialization. By definition, $c_2 < 1$. Put

together, the $w$ term on the right hand side of equation 31 can be ignored close to initialization. Then, $dw/dt \approx (c_1/N)e^\beta$. Dividing this by equation 32 and integrating, we get the conservation equation $w^2/2 - \beta = w_0^2/2 - \beta_0$, so that $w = \sqrt{2(\beta - \beta_0 + w_0^2/2)}$. Substituting this expression for $w$ into equation 32, we have

$$\frac{d\beta}{dt} \approx \frac{c_1 \sqrt{2(\beta - \beta_0 + w_0^2/2)}}{N} e^\beta. \tag{33}$$

Integrating this equation from $\beta = \beta_0$ to $\beta = \infty$ allows us to estimate the time $t_{\text{ICL}}$ it takes to acquire ICL. Performing this integration, we get

$$N\sqrt{2}\Gamma(1/2, w_0^2/2)e^{-\beta_0 + w_0^2/2} \approx I_K(t_{\text{ICL}}) \tag{34}$$

where $\Gamma(.,.)$ is the upper incomplete gamma function and we have defined $I_K(t) \equiv 2\int_0^t c_1(t')dt'$. The subscript $K$ is introduced to highlight $c_1$'s (and thus $I_K$'s) dependence on $K$. In the main text, we examine the case when $w_0 = 0$ so that $t_{\text{ICL}}$ satisfies

$$N\sqrt{2\pi}e^{-\beta_0} \approx I_K(t_{\text{ICL}}). \tag{35}$$

When $-\beta_0 \gg 1$ and $|w_0| \ll 1$, the dynamics can be split into two parts: 1) $w$ converges from its initial value to the nullcline, $dw/dt = 0$, where $w = e^\beta/c_2$, 2) $w$ and $\beta$ gradually increase along this nullcline until $\beta$ is large enough that the exponential dependence in equation 8 leads to abrupt ICL acquisition. The first regime is shorter and $t_{\text{ICL}}$ is dominated by the duration of the latter regime. In the latter regime, since $w = e^\beta/c_2$, we get

$$\frac{d\beta}{dt} = \frac{c_1}{Nc_2} e^{2\beta}. \tag{36}$$

Integrating this equation, we get

$$Ne^{-2\beta_0} \approx I'_K(t_{\text{ICL}}), \quad \text{where } I'_K(t) = \int_0^t \frac{c_1(t')}{c_2(t')} dt'. \tag{37}$$

The $e^{-\beta_0}$ and $e^{-2\beta_0}$ scalings of $t_{\text{ICL}}$ for $|\beta_0| \ll 1$ and $-\beta_0 \gg 1$, respectively, are consistent with those obtained when equations 31 and 32 are numerically integrated beginning from $w_0 = 0$ (Figure A.1). In these numerical simulations, we fix $c_1 = 1/2, c_2 = 1/2$, which is equivalent to setting $z_{\text{MLP}} = 0$ (i.e., the MLP does not contribute to the logit).

## A.4 TRANSIENCE

We examine the influence of applying $L_2$ regularization to the parameters of the attention head. For simplicity, we assume regularization (with $L_2$ regularization parameter $\lambda_w$) is applied only to $w$. Our goal in this section is to show that such a regularization parameter (however small) is necessary to induce transience, and to delineate the values of $\lambda_w$ for which transience is recapitulated in the minimal model. A sufficiently large regularization parameter will of course also affect ICL acquisition (Supplementary Figure A.6). Further analysis could delineate the range of values of $\lambda_w$ for which regularization will have a significant effect on ICL acquisition; this analysis is beyond the scope of this paper.

We write down the dynamics of $w$ *after* ICL is acquired. Re-writing the expression for the loss in equation 22,

$$\mathcal{L} \approx -\left\langle \left(1 + \frac{\eta}{\sqrt{N}}\right) \log \sigma \left(\phi^+ + w\left(\frac{e^\beta - 1}{e^\beta + N - 1} + \frac{\eta\sqrt{N}}{e^\beta + N - 1}\right)\right)\right\rangle_{\eta, \phi^+}. \tag{38}$$

Note that the terms involving $\eta$ inside the logarithm are at most of order $1/\sqrt{N}$. Since $w, \beta \gg 1$ after ICL is acquired, the first term $(e^\beta - 1)/(e^\beta + N - 1) \approx 1$ will be much larger than the term involving $\eta$. This leads to

$$\mathcal{L} \approx -\left\langle \left(1 + \frac{\eta}{\sqrt{N}}\right) \log \sigma \left(\phi^+ + w\right)\right\rangle_{\phi^+}. \tag{39}$$

Since $\langle \eta \rangle = 0$, we have

$$\mathcal{L} \approx \left\langle -\log \sigma \left( \phi^+ + w \right) \right\rangle_{\phi^+}, \tag{40}$$

$$= \left\langle \log \left( 1 + e^{-\left( \phi^+ + w \right)} \right) \right\rangle_{\phi^+}, \tag{41}$$

$$\approx \left\langle e^{-\phi^+} \right\rangle_{\phi^+} e^{-w}. \tag{42}$$

where we have used $e^{-\left( \phi^+ + w \right)} \ll 1$ and $\log(1 + x) \approx x$ for $x \ll 1$. Define $c_3 \equiv \left\langle e^{-\phi^+} \right\rangle_{\phi^+}$, which is our third MLP-specific dynamical variable. $c_3$ will decrease while the MLP memorizes the dataset. Equation 42 implies that, without any regularization, $w$ will continue to increase (albeit at a decreasing rate) and ICL will not be transient.

Introducing an $L_2$ regularization term, we have

$$\mathcal{L} \approx c_3 e^{-w} + \frac{\lambda_w w^2}{2}. \tag{43}$$

The gradient descent dynamics of $w$ is given by

$$\frac{dw}{dt} = c_3 e^{-w} - \lambda_w w. \tag{44}$$

With regularization, once the network sufficiently memorizes the dataset (i.e., $c_3$ is sufficiently small), $w$ reaches a steady state, denoted $w_{\text{tr}}$, which satisfies $w_{\text{tr}} e^{w_{\text{tr}}} = c_3 / \lambda_w$. Re-expressing this relationship in terms of the Lambert W function, we have

$$w_{\text{tr}} \approx W \left( c_3 / \lambda_w \right). \tag{45}$$

Since $c_3 = \left\langle e^{-\phi^+} \right\rangle_{\phi^+}$, we have $c_3 \approx 1$ at initialization. $c_3$ will continue to decrease (however slowly) after ICL is acquired while the loss in equation 43 is minimized. $\lambda_w$ is typically chosen to be a small value. Due to slow memorization (declining $c_3$), $w_{\text{tr}}$ slowly declines while tracking equation 45. ICL fades when $w_{\text{tr}}$ is of order one; since $W(x) = 1$ when $x \approx 3$, i.e., some number of order one, ICL fades when $c_3 \approx \lambda_w$. Our MLP simulations show that $c_3$ can decay further than $10^{-5}$ (Supplementary Figure A.2c).

This analysis suggests that ICL is transient in our minimal model when the network sufficiently memorizes the dataset ($c_3$ is sufficiently small). Importantly, whether ICL is transient or not depends both on the extent to which the network is able to memorize the dataset and how much regularization is applied to each of the network elements. In transformer networks, how regularization may affect each sub-circuit is hard to disentangle. For example, it is possible that the output of the attention head passes through the MLP before producing the final logit. Increasing the regularization on MLP parameters may decrease the rates at which the network memorizes *and* generalizes.

## A.5 THE RELATIONSHIP BETWEEN ICL AND IWL LOSS AFTER ICL ACQUISITION

Recall that IWL is measured on sequences where the item-label pairs are drawn from $\mathcal{D}$ but a copy of the target item is (most likely) not in the sequence. This allows us to define an IWL loss $\mathcal{L}_{\text{IWL}}$, which corresponds to the binary cross-entropy loss of the network on such sequences. Similarly, we define an ICL loss $\mathcal{L}_{\text{ICL}}$, which is the binary cross-entropy loss on input sequences that contain $N$ novel item-label pairs and where the target is one of the $N$ items in the sequence. Equation 45 hints at a relationship between $\mathcal{L}_{\text{ICL}}$ and $\mathcal{L}_{\text{IWL}}$ after ICL is acquired. We now provide a heuristic argument to derive this relationship.

Immediately after ICL is acquired, the ICL loss is small, $\mathcal{L}_{\text{ICL}} \ll 1$. Our arguments in the previous section show that ICL will be transient if the MLP that can sufficiently memorize the dataset and appropriate regularization is applied. Once ICL fades and the network near-perfectly memorizes the dataset, we expect $\mathcal{L}_{\text{IWL}} \ll 1$.

We first consider the case when $\mathcal{L}_{\text{ICL}} \ll 1$, that is, after ICL is acquired but before it fades. ICL acquisition corresponds to $w, \beta \gg 1$. Recall from the independence ansatz that the final logit is $z_{\text{MLP}} + z_{\text{ATT}}$. IWL sequences do not contain a copy of the target in the sequence. Instead, since

$\beta \gg 1$, the target $x_{N+1}$ will pay attention to a random item, say $x_i$, in the input sequence that maximizes the dot product $x_{N+1}.x_i$. Item $i$'s label $\ell_i$ matches the target's label $\ell_c$ with probability half. In the case that it does match, $z_{\text{ATT}} \approx w\ell_i = w\ell_c$. Since the MLP has not yet memorized $\mathcal{D}$, the typical logit from the MLP is small compared to $w$, $|z_{\text{MLP}}| \ll w$. The IWL loss when $\ell_i = \ell_c$ is $-\log \sigma(\ell_c(z_{\text{MLP}} + z_{\text{ATT}})) \approx -\log \sigma(\ell_c(w\ell_i)) = -\log \sigma(w)$, which is negligible since $w \gg 1$. In the alternative case, $\ell_i = -\ell_c$, the IWL loss is $-\log \sigma(w\ell_i\ell_c) = -\log \sigma(-w) = \log(1 + e^w) \approx w$, where we used $w \gg 1$ in the final step. Since $\ell_i = \pm \ell_c$ with equal probability of $1/2$, the expected IWL loss is $\mathcal{L}_{\text{IWL}} \approx w/2$.

On the other hand, ICL sequences do indeed contain a copy of the target in the sequence. In this case, $z_{\text{ATT}} = w\ell_c$. Since $|z_{\text{MLP}}| \ll w$, the ICL loss is $\mathcal{L}_{\text{ICL}} = -\log \sigma(\ell_c(z_{\text{MLP}} + z_{\text{ATT}})) \approx -\log \sigma(\ell_c(w\ell_c)) \approx \log(1 + e^{-w}) \approx e^{-w}$, where we used $\log(1 + x) \approx x$ for $x \ll 1$ in the final step. Putting the expressions for $\mathcal{L}_{\text{IWL}}$ and $\mathcal{L}_{\text{ICL}}$ together, we get

$$\mathcal{L}_{\text{IWL}} \approx -\frac{1}{2} \log \mathcal{L}_{\text{ICL}} \tag{46}$$

after ICL acquisition and before ICL fades away.

We now consider $\mathcal{L}_{\text{IWL}} \ll 1$, that is, after ICL fades away and the network significantly memorizes the dataset. In this case, since $|w|, |\beta| \ll 1$, we have $|z_{\text{ATT}}| \ll |z_{\text{MLP}}|$. The loss on ICL sequences is dominated by the logit produced by the MLP on novel items. Suppose that the typical magnitude of the logit produced by the MLP is $\xi$. The IWL loss is $\mathcal{L}_{\text{IWL}} \approx -\log \sigma(\xi) = \log(1 + e^{-\xi})$. $\mathcal{L}_{\text{IWL}} \ll 1$ implies $\xi \gg 1$ and thus $\mathcal{L}_{\text{IWL}} \approx e^{-\xi}$. For novel items, the MLP will produce either $\pm \xi$ irrespective of the true label of the target. The ICL loss is either negligible or $\log(1 + e^{\xi})$, depending on whether the sign of $z_{\text{MLP}}$ matches the target's label or not. Since $\xi \gg 1$, the ICL loss averaged over these two cases is $\mathcal{L}_{\text{ICL}} \approx \xi/2$. We thus get

$$\mathcal{L}_{\text{ICL}} \approx -\frac{1}{2} \log \mathcal{L}_{\text{IWL}} \tag{47}$$

after ICL fades away and the network significantly memorizes the dataset.

### A.6 SUPPLEMENTARY FIGURES

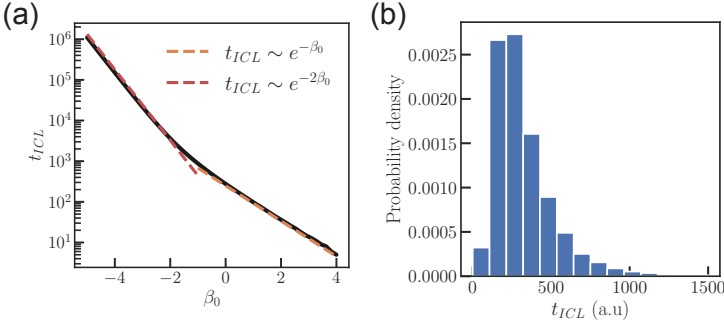

Figure A.1: (a) $t_{\text{ICL}}$ exhibits exponential dependence on the initial values of $\beta_0$. (b) Histogram of $t_{\text{ICL}}$ in the minimal model given normally distributed values of $\beta_0$. The distribution of $t_{\text{ICL}}$ is long-tailed due to its exponential dependence on $\beta_0$.

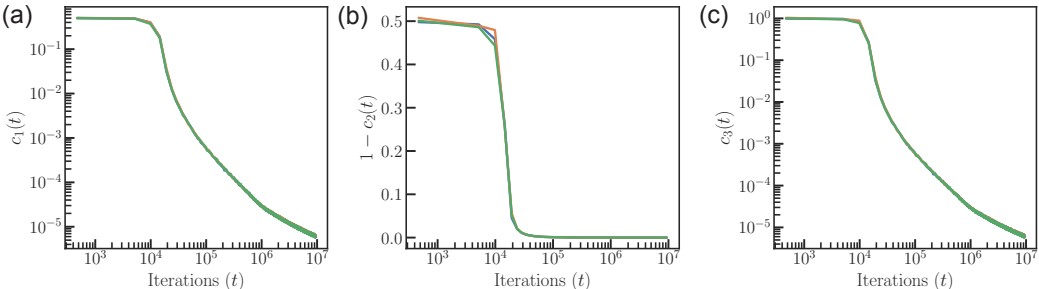

Figure A.2: MLPs were trained to memorize $K$ item-label pairs drawn as in our transformer experiments. We plot the time evolution of three MLP-specific "order parameters". Each plot shows runs from three random seeds. (a) $c_1 \equiv \langle \sigma(-\phi^+) \rangle_{\phi^+}$. (b) $c_2 \equiv 1 - \langle \sigma(-\phi^+)^2 \rangle_{\phi^+} / c_1$ (c) $c_3(t) = \left\langle e^{-\phi^+} \right\rangle_{\phi^+}$. The variability across seeds is negligible.

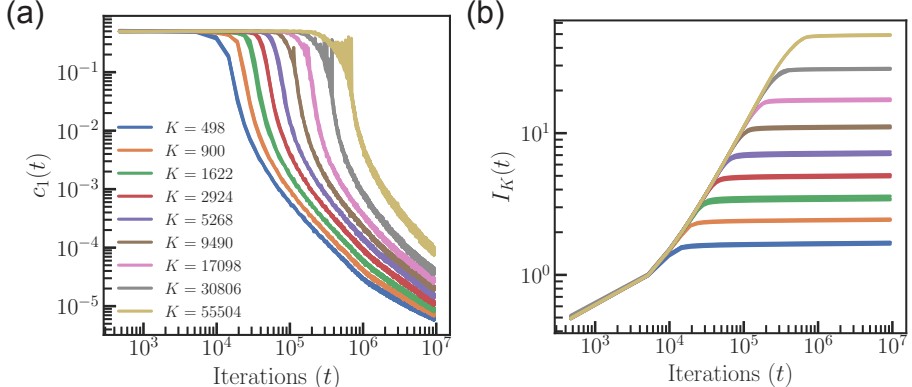

Figure A.3: (a) MLPs were trained to memorize $K$ item-label pairs drawn as in our transformer experiments. We plot $c_1 = \langle e^{-\phi^+} \rangle_{\phi^+}$ for different $K$. (b) Time evolution of $I_K(t) = 2 \int_0^t c_1(t') dt'$ (upto a constant prefactor) obtained by numerically integrating the curves in panel (a).

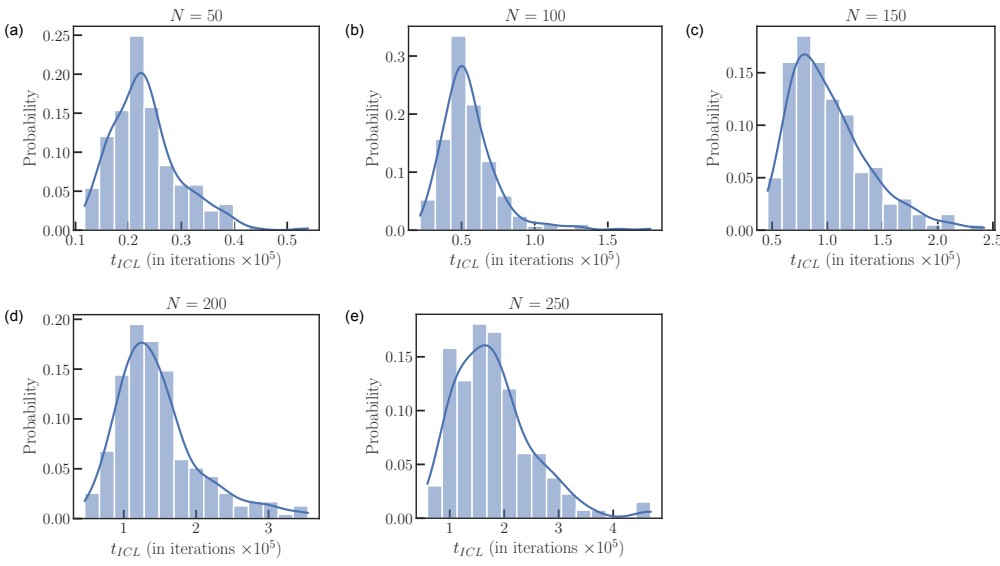

Figure A.4: For each $N$, we trained the model across at least 130 seeds and show that histograms of $t_{\text{ICL}}$ are long-tailed.

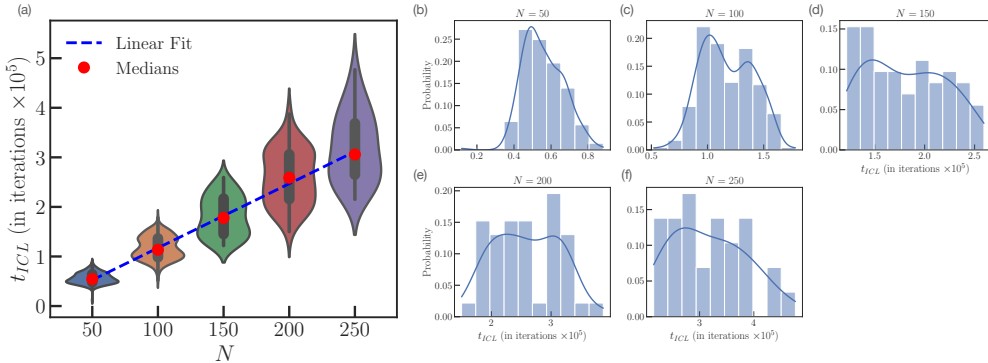

Figure A.5: For a two-layer transformer, we replicate the finding that $t_{\text{ICL}}$ scales linearly with context length $N$. We trained the model across at least 29 seeds for each $N$ and show that histograms of $t_{\text{ICL}}$ are broadly long-tailed.

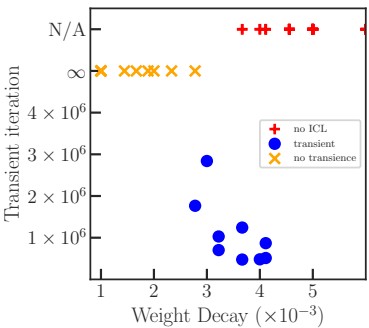

Figure A.6: To explore the impact of $L_2$ regularization on ICL transience, we trained models while varying the weight decay $\lambda_w$ applied on the self-attention layer. For each run, we compute the iteration at which we observe transience, which we define as the first iteration after the model has attained at least 99% ICL accuracy and after the model's ICL accuracy has declined to 90%. The red plus signs indicate runs in which the model never attains the ICL solution (never attains at least 99% ICL accuracy). The yellow crosses indicate runs in which the model does attain the ICL solution but we do not observe transience, as the time to observe transience is too long. The blue dots are runs where we do observe transience, showing that increasing $\lambda_w$ induces faster transience.

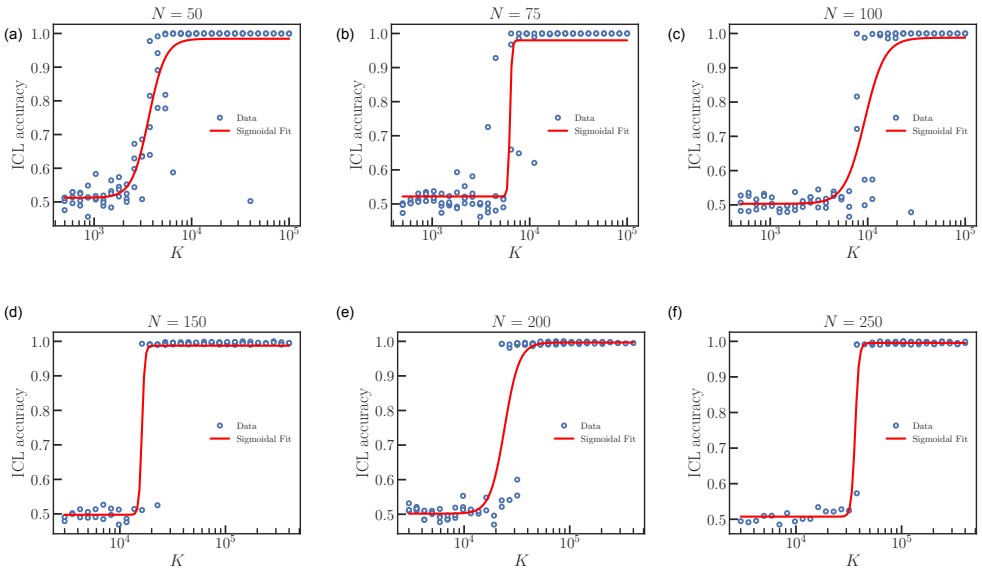

Figure A.7: ICL performance is measured for various $N$ and $K$. For all $N$, we see a sharp transition from memorization to generalization as $K$ increases. We fit a sigmoid to the data to determine the critical task diversity threshold $K^*$ for each $N$.

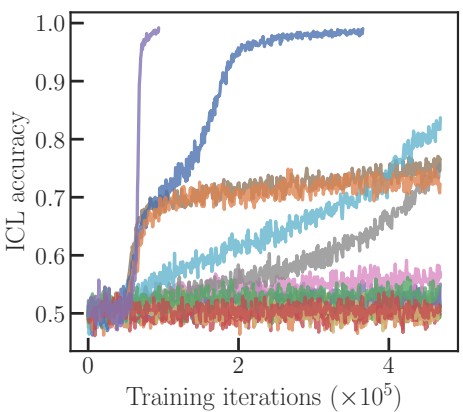

Figure A.8: ICL performance curves (for different random number seeds) for models trained on our modified $\mathcal{D}'$, where every sequence contains exactly $N/2$ tokens with $+1$ and $-1$ labels for $K = 31336$. We see novel intermediate suboptimal solutions, and many of the runs never acquire ICL.

