# OpenReview forum: "Differential learning kinetics govern the transition from memorization to generalization during in-context learning"
_ICLR.cc/2025/Conference — ICLR 2025 Spotlight_

### Official Review · Reviewer_SXqf · 2024-10-24

**Soundness:** 4
**Presentation:** 4
**Contribution:** 3
**Rating:** 8
**Confidence:** 4

**Summary:**

The paper provides a mechanistic interpretability analysis of ICL and IWL, explaining observed phenomena from the literature and presenting testable predictions for other spacing laws. The authors hypothesize and experimentally demonstrate that the sub-circuits responsible for memorization and generalization can largely be considered independent.

**Strengths:**

First of all, congratulations on your ICLR submission! I’d like to commend the author for this excellent piece of research. Below is my review of the work:

**Originality**
The paper is bringing a novel methodology to study ICL and IWL, and understand the trade of between the generalisation and memorisation.

**Quality**
The paper is of general great quality
- The paper effectively acknowledges its limitations.
- The experimental results are well-detailed and thoughtfully motivated.
- The findings demonstrate that the predictions made using the smaller model hold up in a slightly more general model used to study ICL and IWL.

**Clarity**
The paper is very well-structured, making it easy to follow. The figures are clear, informative, and complement the content effectively.

**Significance**
This work is highly relevant to the research community and plays a crucial role in improving the interpretability of neural networks. I would like to highlight its significance, particularly for the neuroscience community, where there is potential for fascinating cross-disciplinary ideas, as exemplified in this paper see . These paper explore bi level optimisation as a mechanism for effective learning and generalisation [1,2].

[1] Jensen, K.T., Hennequin, G. and Mattar, M.G., 2024. A recurrent network model of planning explains hippocampal replay and human behavior. Nature Neuroscience, pp.1-9.
[2] Murray, J.M. and Escola, G.S., 2020. Remembrance of things practiced with fast and slow learning in cortical and subcortical pathways. Nature Communications, 11(1), p.6441.

**Weaknesses:**

**Originality**
The topic is highly relevant and aligns with current research trends, yet it still offers an intriguing and fresh perspective.

**Quality**
A more detailed discussion of the limitations and assumptions would enrich the paper. Additionally, delving deeper into the comparison with capacity theory could provide further insights (see question below).

**Clarity**
Suggestion: Including a small diagram of the model would improve readability, especially considering its compact size.
Could the results be validated in a more step-by-step manner? This would clarify that the approximations are sound and the rationale behind them, potentially expanding on this in the appendix.
Providing full derivations in the appendix would enhance completeness and transparency.

**Significance**
While the authors have noted the need for validation with larger models, doing so would strengthen the paper, despite the acknowledged material constraints.

**Questions:**

**Questions**

- In previous works that observed the transient nature of the network, were weight regularizers applied?
- Could there be an alternative explanation beyond the dynamical and capacity arguments?
- For future work, it would be interesting to explore the dynamic effects under capacity limitations. Do you have any intuition about the phenomena that might arise, considering that both could occur in real-world models?
- How do these insights from ICL and IWL models compare with findings at scale in previous literature? Do you expect some of your results to hold at scale?

I hope this review is helpful for the further development of the paper. I encourage the author to continue this research and incorporate the feedback provided.

---

> ### Author Response · Authors · 2024-11-27
> **Responses to Reviewer SXqf**
>
> We thank the reviewer for their review of our article. Here are our responses to their comments:
>
> > A more detailed discussion of the limitations and assumptions would enrich the paper... Providing full derivations in the appendix would enhance completeness and transparency.
>
> We have added a new appendix that provides additional details on the derivations and assumptions made in the main text. We have also clarified our assumptions in the main text based on our discussions with the reviewers.
>
> > Suggestion: Including a small diagram of the model would improve readability, especially considering its compact size. Could the results be validated in a more step-by-step manner?
>
> We provided a diagram of the model in Figure 2A. Please let us know if there are any additional details that you would like to see in the diagram, or if you have other concerns about the validation of our results.
>
> > In previous works that observed the transient nature of the network, were weight regularizers applied?
>
> In the work of Singh et al. [2], the authors applied weight decay to the self-attention layers of the transformer which can cause ICL to be transient. Puzzlingly, without any L2 regularization, ICL eventually decays when the model is trained for a sufficiently large number of iterations. We added this observation to the main text and clarified how our theoretical results relate to their results. Singh et al also observed that as L2 regularization is applied, ICL transience is mitigated. Furthermore, as L2 regularization is increased even further (above $10^ {-4}$),  ICL transience returns, but the network also never learns the IWL solution. They also found that when weight decay is applied only to self-attention layers, ICL is still transient but when weight decay is applied only to MLP layers, ICL is mitigated.
>
> > Could there be an alternative explanation beyond the dynamical and capacity arguments? For future work, it would be interesting to explore the dynamic effects under capacity limitations. Do you have any intuition about the phenomena that might arise, considering that both could occur in real-world models?
>
> We have provided additional experimental results computing the MLP order parameters using 10 times more iterations and observe a regime shift in MLP memorization kinetics above $5\times 10^8$ iterations, such that the power law no longer holds. This hints at some dynamical effects that can be explored in future work. Another avenue to explore is how the optimizer choice can affect the solution found by the model. Intuitively, we think that Adam can escape the flat regions of the loss landscape more easily than SGD, which can reduce the time to ICL and allow the model to exhibit learning dynamics that we do not observe with SGD.
>
> > While the authors have noted the need for validation with larger models, doing so would strengthen the paper, despite the acknowledged material constraints... How do these insights from ICL and IWL models compare with findings at scale in previous literature? Do you expect some of your results to hold at scale?
>
> We refer to the work of Kirsch et al. [1] who observed bimodality in the transition from memorization to generalization in a multi-layer transformer model as data diversity increases. Furthermore, we refer to the work of Singh et al. [2] on the transience of ICL performance in multi-layer transformers on an in-context classification task similar to ours, showing that these phenomena also hold at scale.
> 1. Kirsch, L., Harrison, J., Sohl-Dickstein, J., & Metz, L. (2022). General-purpose in-context learning by meta-learning transformers. arXiv preprint arXiv: 2212.04458.
> 2. Singh, A., Chan, S., Moskovitz, T., Grant, E., Saxe, A., & Hill, F. (2024). The transient nature of emergent in-context learning in transformers. Advances in Neural Information Processing Systems, 36.

---

> > ### Comment · Reviewer_SXqf · 2024-11-29
> >
> > I commend the authors for their clear explanations and thorough responses to the questions raised. The discussion of the work's relevance, particularly in relation to Singh et al. (2024), was especially insightful. The updates to the paper are valuable, and I will maintain my positive rating of 8 for this work.

---

### Official Review · Reviewer_4B2U · 2024-11-03

**Soundness:** 4
**Presentation:** 4
**Contribution:** 4
**Rating:** 10
**Confidence:** 4

**Summary:**

The paper studies the transition behavior of in-context learning (ICL) and in-weight learning (IWL) in single-layer, single-head, nonlinear Transformer, trained on a variant of the binary labelling task introduced by Chan (2022). The paper then reduces this setting to an effective, “minimal” model that treats ICL and IWL as separate terms, leading to a pair of coupled scalar ODEs stemming from gradient flow on a data-averaged loss function. The minimal model is able to qualitatively explain empirical phenomena in the literature. Importantly, for the setting considered, the provided explanations match quantitatively, and with high precision.

In particular, the model explains the following empirical phenomena

1. Sharp acquisition of ICL w.r.t. task diversity (modelled by number of training samples $K$), with bimodal IWL-ICL distribution at the threshold K\* (Kirsch, 2022\)
2. Long-tailed distribution of ICL acquisition times (Reddy, 2023\)
3. Transience of ICL (Singh, 2023\)
4. (somewhat) The dependence on data statistics, in the paper modelled as label inbalance $n\_+/N$

In addition, the model makes two predictions: 1) That acquisition time $t_{ICL}$ scales linearly with context length $N$, and that "task diversity" $K^\star$ where ICL and IWL coexist depends on context length $N$ with a power law $ K^* \sim N^\{\nu=0.72\}$.

### Recommendation
The explanatory power, originality, and simplicity of the model are striking and well outweigh my criticisms. Moreover, the paper is highly relevant for the current discussion on in-context learning. I recommend acceptance and a prominent highlight at the conference.

**Strengths:**

- The modelling approach is plausible and original, and the power despite its simplicity of the minimal model is striking.

**Weaknesses:**

(more important points first)

- While the model qualitatively explains the literature as stated above, quantitative connections are missing. I understand that some quantitative predictions may be contingent on the one-layer-one-head architecture and task, but others may be transferable, such as the two predictions mentioned in the Summary. I would find it interesting to evaluate these on a naturalistic task. Two examples come to mind: The path-finding task introduces in (Brinkmann et al., 2024) and possibly also a lightweight NLP setting.
- It is unclear to me how strong the dependence is in terms of data statistics. In particular, is there an intuition why the edge case of exactly $N/2$ positive $\ell=\+1$ tokens is so different?
- It took me a while to understand the relation between the two losses at the end of Section 3.7. It would help the reader to find details, possibly in the Appendix.
- Why does $\\beta=\\infty$ (or even just a large value) in Eq. (12) signify ICL dominance? I see this for $w$, but not for $\\beta$.
- The claim that $c\_1(t)$ has a power-law up to $K=10^5$ is not supported by Fig. A.3. The claim needs to be weakened or additional simulations are required.
- The title of the paper, “differential learning kinetics”, that also is used to label the distinction in Fig. 1, is not picked up much in the main text, and in my opinion also not very descriptive overall. I feel the paper would benefit from connecting the theory to this intuition that the authors developed.
- In Fig. 1 and 2: At what time is $K^*$ evaluated?
- No code is published, limiting transparency and reproduciblity.

#### **Minor**

- The paper states in line 376
> While previous work has shown that such skewed data distributions favor ICL acquisition (Chan et al. (2022)), our theory suggests that such data dependencies can be examined by simply measuring $I\_K$.

  As I understand these findings are compatible, the wording here to me however makes it unclear whether they are or not.
- It is not clear how good of a proxy $K$ is for data diversity, generally, so this caveat should be mentioned.
- Sometimes $\rangle\_{\\phi\_+}$ and sometimes  $\rangle\_{\\phi\_+ \\sim P\_+}$ are used, that was slightly confusing to me. Also, $P_\+$ and $P(n_+)$ could be differentiated more clearly notationally.
- In line 413 it says
> Equation 16 implies that $w$ after ICL acquisition tracks $w_{tr}$...

I assume $w_{tr}$ stems from gradient descent on Eq. (15), but this and why it bears this particular name is not clear.
- Data diversity is modelled by $K$, I would indicate this in Fig. 1 on the y-axis
- In line 361, the case $-\beta\_0 \gg 1$ is discussed, and then a conclusion is being made on long-tailedness. I believe that the results show that long-tailedness should hold for both $-\beta\_0 \\gg 1$ and $-\\beta\_0 \\ll 1$, but the wording here made it seem like it only applies to the former.
- Just before Section 3.3, the paper states
> …three phenomena of interest, the transition from memorization to generalization, abrupt ICL learning and ICL transience,...

  of which the first and second seem identical, and the (Reddy 2023\) long-tailedness property does not appear.
- The Gaussian on the y-axis s Fig. 6c is not referenced.

**Questions:**

I've expressed my questions in the Weaknesses section above.

---

> ### Author Response · Authors · 2024-11-27
> **Response #1**
>
> We thank the reviewer for the detailed comments. Here are our specific responses:
>
> > While the model qualitatively explains the literature as stated above, quantitative connections are missing. I understand that some quantitative predictions may be contingent on the one-layer-one-head architecture and task, but others may be transferable, such as the two predictions mentioned in the Summary. I would find it interesting to evaluate these on a naturalistic task. Two examples come to mind: The path-finding task introduces in (Brinkmann et al., 2024) and possibly also a lightweight NLP setting.
>
> We could not implement a different task, but we recapitulated our prediction of a linear scaling of $t_{\text{ICL}}$ against context length $N$ in a two-layer transformer (Supp Fig A.5 in updated version). Testing the other predictions required many more simulations with a larger model than we could perform.
>
> > It is unclear to me how strong the dependence is in terms of data statistics. In particular, is there an intuition why the edge case of exactly positive tokens is so different?
>
> We now expand on the intuition in lines 326-330. Pasting it here: "Intuitively, when $n_+$ is binomially distributed, the network learns that the target's label is more likely to be a $+1$ if there are more $+1$ labels than $-1$ labels in the context. This bias, however small, pushes $w$ to a positive value and leads the network into the ICL basin. When the numbers of $+1$ and $-1$ labels are forced to be equal, $w$ could flow into the basin that leads to ICL or flow to an alternative potentially suboptimal solution."
>
> > It took me a while to understand the relation between the two losses at the end of Section 3.7. It would help the reader to find details, possibly in the Appendix.
>
> We wrote a new section in the Appendix expanding on the relation (Section A.5).
>
> > Why does (or even just a large value) in Eq. (12) signify ICL dominance? I see this for , but not for .
>
> After $\beta$ is of order one, $\beta$ increases exponentially (see equation 33 for example). The difference in the time it takes $\beta$ to get close to 1 and to get to $\infty$ is not much, so we might as well take the integral to $\infty$ rather than some arbitrarily defined threshold.
>
> > The claim that  has a power-law up to  is not supported by Fig. A.3. The claim needs to be weakened or additional simulations are required.
>
> We reran the simulations for 10x more iterations and we found that the power-law in fact does not remain for longer $t$. The integral $I_K(t)$ still saturates so the rest of our results still hold, but the claim that there is a power-law is not true. We modified our text to reflect the results from these simulations. Note that when we re-did our simulations with 10x more iterations, the measured exponent showed a small change: $\nu =0.70$ rather than $\nu = 0.72$.
>
> > The title of the paper, “differential learning kinetics”, that also is used to label the distinction in Fig. 1, is not picked up much in the main text, and in my opinion also not very descriptive overall. I feel the paper would benefit from connecting the theory to this intuition that the authors developed.
>
> Due to the addition of the appendix, we had more room to expand on this point. We now discuss the "differential learning kinetics" vs "capacity-constrained" mechanisms more extensively in the main text (Section 3.6)

---

> > ### Author Response · Authors · 2024-11-27
> > **Response #2**
> >
> > > In Fig. 1 and 2: At what time is evaluated?
> >
> > Maybe the reviewer could clarify what they meant? We were not sure.
> >
> > > No code is published, limiting transparency and reproduciblity.
> >
> > We hope to upload our code once the anonymity restrictions are lifted.
> >
> > Other minor comments: We made numerous minor changes to improve the presentation and to address the minor comments raised by Reviewer 4B2U. Note that none of the results have changed (except for the tail behavior of $c_1$ ad the small shift in the exponent $\nu$ noted in the previous comment).

---

> > > ### Comment · Reviewer_4B2U · 2024-11-30
> > >
> > > Thank you for your response! The added details in the appendix make it much easier to understand the derivations in the manuscript. I also appreciate the additional simulations carried out, in particular it is reassuring to see results for the two-layer transformer. Still, the biggest open question is to what extent the quantitative predictions of the paper at least somewhat carry over to more complicated settings, but I still don't think this impacts my evaluation of this work.
> > >
> > > > In Fig. 1 and 2: At what time is evaluated?
> > >
> > > > Maybe the reviewer could clarify what they meant? We were not sure.
> > >
> > > My apologies about the confused phrasing in my review, I was asking what time in training the relevant panels in Figures _2 and 3_ show.
> > >
> > > > We hope to upload our code once the anonymity restrictions are lifted.
> > >
> > > It is possible to upload code with the supplementary or provide an anonymized link.

---

### Official Review · Reviewer_EVew · 2024-11-03

**Soundness:** 1
**Presentation:** 2
**Contribution:** 1
**Rating:** 3
**Confidence:** 5

**Summary:**

This paper presents a small transformer model performing a synthetic task aimed at separating in-context and in-weights learning. A surrogate model which is amenable to analysis is also presented. The paper aims to combine the analysis of this ``minimal model'' and the empirical analysis of the original transformer to understand under what conditions in-context learning emerges. The training dynamics of the minimal model are characterised and used to analyse the loss landscape of the in-context pathway of the model. Further, a number of phenomena of transformers performing in-context learning are considered, such as the transient nature of this capability.

**Strengths:**

# Originality
To my knowledge this is a new combination of ideas from the literature. The tasks and network structure similar to the work of Chan et. al. (2022) is combined with the analysis of the loss landscape and loss dynamics in various high-dimensional limits (while it is not mentioned in the work, the analysis has D and K very larger - the input dimension and number of datapoints - which constitutes the well known thermodynamic limit). By working in this limit much of the dynamics of training become tractable and this is exploited here.

# Quality
The tasks and model used here is well motivated and clearly relevant to in-context learning and useful for the aims of the work. The hypotheses or motivation of the work is clear.

#  Clarity
Overall use of language in this work is clear and it is well written. Figures are neat and legible.

# Significance
This work touches on a number of established phenomena on in-context learning but also introduces new phenomena which would be useful for further research. The model itself could likely be built upon in further work.

**Weaknesses:**

# Quality
The work is vague and lacks the necessary degree of rigor. The fact that no full derivations are presented even in the appendix is already grounds for rejection in my opinion. I recommend the authors provide a more thorough presentation of the derivations in this work.

This somewhat casual approach to the mathematics means that some very important simplify assumptions are not described accurately and in some case the work changes assumptions or conditions to suit whatever point it is currently trying to make. The most clear example of this is the dimension of the input D. It is foundational to the analysis here that D is large enough that the key-query matrix will be diagonal. This appears to me to be a huge simplification, worthy of clear mention. But the work fails to even acknowledge this in Section 3.2 when describing the model, instead referencing the fact indirectly and terming it an ``independence ansatz''. This fact of needing infinite D is only brought up in Section 3.3. This lack of clear assumptions and formal definition results in the jump from Equation 3 to Equation 4 being unclear. A similar case can be made for the implicit simplicity of the setup of binary classification itself - which is necessary for the analysis which relies of a single variable specifying the probability of the  MLP being correct ($P^+$). Thus, the jump from Equation 4 to the equation after the sentence  "since the two labels are symmetric the average loss $\mathcal{L}$ can be written as" (which again is not actually shown) is also unclear and missing multiple steps. Another example is the treatment of $N$ right before Equation 5, where it is never mentioned that N follows a distribution before this point, but then it is just defined as being approximated by a normal distribution and the loss equation updated again without derivation or explanation. This happens throughout the work and needs to be correct, for example before Equation 6 it is merely stated "From Equation 5 a few steps of simplification leads to...". I think I make my point but for clarity I will point out some quicker examples:
- Lines 297 to 302: The conclusion of the loss from the MLP separating from the in-context pathway seems true by construction and if I am not mistaken is essentially the independence ansatz. I'm not certain why here it is being phrased as an emergent phenomenon.
- Figure 4: The landscape depends on the probability of the MLP being correct but this is omitted entirely.
- Lines 329 to 330: it says "if the MLP is unable to perfectly memorize the K item-label pairs" but on lines 225 and 226 it is said that "all the information required by the MLP to memorize the label of the target is present in $t_{N+1}$". These are contradictory statements. It seems the second sentence is an assumption which is used in the derivation (or else you cannot summarize the MLP contribution just with $P^+$ - it is difficult to tell without the derivation). This would mean that you cannot interpret the model in the case of the MLP not being able to learn.
- Lines 336 to 338: The ultimate conclusion of this section is that the model behaviour depends on the initial values of $\omega$ and $\beta$ (less the MLP cannot learn the dataset in which case in-context learn will clean up the remaining loss as best it can). Given this conclusion I am confused what the point of this section is given that the condition of $\omega, \beta >> 1$ resulting in ICL is already noted from Equation 3. It's possible I am missing some nuance around this point but without derivations it is difficult to say.
- Two Taylor expansions are used in this work with almost no mention of what sorts of terms or information is lost but this approximation.
- Lines 355 and 356: It is not clear at all that $c_1(t),c_2(t) \approx 1/2$ here. This is likely due to the information treatment and lack of explanation of $I_k(t)$.
- Lines 369 to 372: The relationship between $c_1$ and time is not clearly discussed and $c_1$ in it's own right is not clear explained. Why is the mean of the sigmoid of the negative of the MLP output for the positive class useful? Why would it's integral over time diverge? Surely for a fixed dataset the MLP output stabilises? Is this integral then just a repeated sum of this stable mean value? Why does the divergence of $I_K(t)$ then imply ICL will occur?
- For Section 3.7: How standard is it to regularize only the in-context pathway? Seems like a fairly trivial result that if only this pathway is regularized then it will disappear.
- For Section 3.7: Since the dynamics are only considered once ICL is acquired, are the results of this section only valid if L2 regularization is applied once ICL is acquired? Since Equation 5 is not derived with L2 regularization I don't see how it can be just dropped in here and claimed that it ``applies throughout training'' and then the regularization term is just appended. I find it hard to believe that if the regularization term is present throughout training then ICL will emerge in the same manner as the  model which is used to option Equation 5.
- Line 402: it is mentioned that the added regularization may be explicit or implicit. What justifies the conclusion that any sort of regularization will hold in a similar manner (or  is that not the implication of this statement?).  What sort of implicit regularization will only apply a weight normalization to t he in-context pathway?

Unfortunately, given my confusion with the derivation due to the lack of rigor and derivations it is difficult to even assess Section 4. Many of the phenomena which are considered in this section are either due to fairly vague conclusions drawn on equations without clear meaning or seem to be true by construction (again having sufficiently large D that the key-query matrix is diagonal is a large simplification not given sufficient attention). Thus, the conclusions and insights of this section seem premature.

It is possible I am missing something in the research - indeed it is possible this research is very impressive. I also appreciate that theory is difficult and requires simplifying assumptions, especially to get to grips with something as complex as in-context learning in transformers. But I am certain that the presentation of this research is significantly lacking for the current version.

**Questions:**

- What is the $P^+$ value used to plot Figure 4 and how does it affect the loss landscape?
- Figure 5: the power law is presented as being between $K$ and $I_K(t)$ but then Figure 6a presents it as being between $N$ and $K^*$. How do these two representations relate?

---

> ### Author Response · Authors · 2024-11-25
> **Revised version of the manuscript with details of the minimal model in the Appendix**
>
> A new version of the manuscript has been uploaded. Please see the official comment above. We hope that this update will give you some extra time to read through the appendix until we revise the manuscript further and prepare a more detailed response.

---

> > ### Comment · Reviewer_EVew · 2024-11-25
> > **Noting Revision**
> >
> > Good day Authors
> >
> > I note that you have submitted a revised version of the paper with a much extended appendix. Given its length and the quantity of changes I cannot guarantee a revised score due to this revision in time for the end of the discussion period. I will however try my best. I would like to note that many of my concerns are in the main text, and as far as I can tell this remains relatively the same (I have just skimmed the revised version now), is this correct? I look forward to the responses to my comments.
> >
> > Best \
> > Reviewer EVew

---

> > > ### Author Response · Authors · 2024-11-25
> > > **Clarifying revisions**
> > >
> > > Yes, that's correct -- we added an extended version of the analysis in the main text (the results are all the same) to address the general concern that derivation steps are missing. We hope to get back to you soon with a more detailed response and additional revisions of the main text.

---

> > > > ### Author Response · Authors · 2024-11-25
> > > > **Response #1**
> > > >
> > > > We thank the reviewer for the detailed comments. Here are more specific responses to the concerns raised.
> > > >
> > > > > This somewhat casual approach to the mathematics means that some very important simplify assumptions are not described accurately and in some case the work changes assumptions or conditions to suit whatever point it is currently trying to make.
> > > >
> > > > The new appendix fills out the details of the derivations and adds additional statements to clarify the assumptions made in the main text.
> > > >
> > > > >The most clear example of this is the dimension of the input D. It is foundational to the analysis here that D is large enough that the key-query matrix will be diagonal. This appears to me to be a huge simplification, worthy of clear mention. But the work fails to even acknowledge this in Section 3.2 when describing the model, instead referencing the fact indirectly and terming it an ``independence ansatz''. This fact of needing infinite D is only brought up in Section 3.3. This lack of clear assumptions and formal definition results in the jump from Equation 3 to Equation 4 being unclear.
> > > >
> > > > That the key-query matrix is diagonal is an assumption (our interaction strength ansatz) as stated in lines 213 to 222, and we do not claim that it follows from a large D assumption. We stress that our minimal model is *not* derived -- it is a phenomenological model meant to capture the phenomenology displayed by the full model. One of our main results is that such a highly simplified model can also capture the phenomenology displayed by the transformer model. It's simplicity allows us to make progress on the theory front. We now highlight this point in lines 608-626.
> > > >
> > > > Note that D is a finite number (D=63, which is the same value used when training the transformer model) in our simulations with the minimal model shown in Figure 3. The D infinity limit is taken only when we analytically derive the expression for the loss landscape. We hope that the appendix will make the jump from Equation 3 to Equation 4 clearer (see lines 635-647).
> > > >
> > > > > A similar case can be made for the implicit simplicity of the setup of binary classification itself - which is necessary for the analysis which relies of a single variable specifying the probability of the MLP being correct (P+). Thus, the jump from Equation 4 to the equation after the sentence "since the two labels are symmetric the average loss L can be written as" (which again is not actually shown) is also unclear and missing multiple steps.
> > > >
> > > > We now include more steps and clarifying statements explaining the jump from Equation 4 to Equation 5 (see lines 660-690).
> > > >
> > > > > Another example is the treatment of N right before Equation 5, where it is never mentioned that N follows a distribution before this point, but then it is just defined as being approximated by a normal distribution and the loss equation updated again without derivation or explanation. This happens throughout the work and needs to be correct, for example before Equation 6 it is merely stated "From Equation 5 a few steps of simplification leads to...".
> > > >
> > > > $N$ is not drawn from a distribution -- it is kept fixed throughout. However, the number of +1 labels (denoted $n_+$) is a binomial distribution with $p = 1/2$ because the item-label pairs in a sequence are drawn randomly from the dataset D (which contains equal numbers of items with +1 and -1 labels).
> > > >
> > > > We expand on the steps following Equation 5 in lines 694-720.

---

> > > > > ### Author Response · Authors · 2024-11-25
> > > > > **Response #2**
> > > > >
> > > > > > Lines 297 to 302: The conclusion of the loss from the MLP separating from the in-context pathway seems true by construction and if I am not mistaken is essentially the independence ansatz. I'm not certain why here it is being phrased as an emergent phenomenon.
> > > > >
> > > > > This is not true by construction. If ICL were to happen gradually, then the decrease in loss will slow down how quickly the MLP memorizes the dataset. However, since ICL occurs abruptly, we can split the analysis by examining separately what happens before ICL acquisition and after ICL acquisition. In the former phase, the ICL dynamics near initialization does not affect IWL as all the parameters are small (of order $w/N$).
> > > > >
> > > > > > Figure 4: The landscape depends on the probability of the MLP being correct but this is omitted entirely.
> > > > >
> > > > > Thanks for the catch – we now clarify in the Figure 4 caption that the landscape corresponds to the case when the MLP logits are zero. The landscape in general does indeed depend on P^+. Our goal in Figure 4 is to show that the ICL loss landscape is nearly flat at initialization.
> > > > >
> > > > > > Lines 329 to 330: it says "if the MLP is unable to perfectly memorize the K item-label pairs" but on lines 225 and 226 it is said that "all the information required by the MLP to memorize the label of the target is present in
> > > > > ". These are contradictory statements. It seems the second sentence is an assumption which is used in the derivation (or else you cannot summarize the MLP contribution just with, it is difficult to tell without the derivation). This would mean that you cannot interpret the model in the case of the MLP not being able to learn.
> > > > >
> > > > > The MLP does indeed receive all the information necessary to memorize, but it’s still possible that the MLP is not sufficiently complex to perfectly memorize the K item-label pairs. Some clarification on why these are contradictory statements will help us modify the text accordingly.
> > > > >
> > > > > The second sentence is not an assumption – the MLP in a standard transformer model receives the input from the target item through the residual stream.
> > > > >
> > > > > > Lines 336 to 338: The ultimate conclusion of this section is that the model behaviour depends on the initial values of and (less the MLP cannot learn the dataset in which case in-context learn will clean up the remaining loss as best it can). Given this conclusion I am confused what the point of this section is given that the condition of resulting in ICL is already noted from Equation 3. It's possible I am missing some nuance around this point but without derivations it is difficult to say. Two Taylor expansions are used in this work with almost no mention of what sorts of terms or information is lost but this approximation.
> > > > >
> > > > > In lines 327-338, we argue that ICL is acquired irrespective of the initial values of $w$ and $\beta$. However, this depends on the fact that $B(n_+)$ is a binomial distribution. If one were to instead force an equal number of +1 and -1 labels in the input sequence ($N/2$ of each label), then our analysis in lines 331-338 shows that the behavior of the model near initialization is qualitatively different. That is, $w=0, \beta = 0$ saddle point and where the model converges to depends on where you start. We validate these predictions later in the manuscript using our one-layer transformer model (lines 499-507 and Figure A.6).
> > > > >
> > > > > We hope the extended derivations in the appendix explaining the steps from equations 3 to 5 (lines 694-720) will help clarify the concern about Taylor expansions.
> > > > >
> > > > > >Lines 355 and 356: It is not clear at all that c1,c2=1/2 here. This is likely due to the information treatment and lack of explanation of Ik(t)
> > > > >
> > > > > When the MLP is absent or has barely memorized the dataset, we get $c_1, c_2 = 1/2$ from their definitions in line 292. If the MLP has not memorized, then the logit it will produce is $\approx 0$. Since $\sigma(0) = ½$, this implies $c_1 = ½$. Similarly, $c_2 = ½$ from its definition. $I_K(t)$ is defined in Equation 12.

---

> > > > > > ### Author Response · Authors · 2024-11-25
> > > > > > **Response #3**
> > > > > >
> > > > > > > For Section 3.7: How standard is it to regularize only the in-context pathway? Seems like a fairly trivial result that if only this pathway is regularized then it will disappear.
> > > > > >
> > > > > > Our goal in Section 3.7 is to explain the results of Figure 3c, where we were able to recapitulate transience but only if the in-context pathway is regularized. We were able to explain this using our analytical framework, which also leads to a specific prediction for how the ICL loss and IWL loss should be related after ICL is acquired. We clarify the motivation behind Section 3.7 in the Appendix (we will update the main text soon to reflect the addition of the appendix).
> > > > > >
> > > > > > > For Section 3.7: Since the dynamics are only considered once ICL is acquired, are the results of this section only valid if L2 regularization is applied once ICL is acquired? Since Equation 5 is not derived with L2 regularization I don't see how it can be just dropped in here and claimed that it ``applies throughout training'' and then the regularization term is just appended. I find it hard to believe that if the regularization term is present throughout training then ICL will emerge in the same manner as the model which is used to option Equation 5.  Line 402: it is mentioned that the added regularization may be explicit or implicit. What justifies the conclusion that any sort of regularization will hold in a similar manner (or is that not the implication of this statement?). What sort of implicit regularization will only apply a weight normalization to t he in-context pathway?
> > > > > >
> > > > > > The empirical results showing transience in Figure 2e with the one-layer transformer model and in Figure 3c with the minimal model both show ICL acquisition much like in the case when regularization is not applied to the in-context pathway (Figure 2d and Figure 3b). Regularization empirically does not change how ICL is acquired (though of course if it is set too high, the network parameters will not budge).
> > > > > >
> > > > > > However, the L2 regularization of the in-context pathway makes ICL fade away with fewer # of training iterations. The original paper that first characterized transience (Singh et al, Neurips 2023) trained the transformers for orders of magnitude longer (> 10^7 # of iters) than us; unfortunately, we do not have the compute to train our models for 10^7 iterations.
> > > > > >
> > > > > > > Unfortunately, given my confusion with the derivation due to the lack of rigor and derivations it is difficult to even assess Section 4. Many of the phenomena which are considered in this section are either due to fairly vague conclusions drawn on equations without clear meaning or seem to be true by construction (again having sufficiently large D that the key-query matrix is diagonal is a large simplification not given sufficient attention). Thus, the conclusions and insights of this section seem premature.
> > > > > >
> > > > > > > It is possible I am missing something in the research - indeed it is possible this research is very impressive. I also appreciate that theory is difficult and requires simplifying assumptions, especially to get to grips with something as complex as in-context learning in transformers. But I am certain that the presentation of this research is significantly lacking for the current version.
> > > > > >
> > > > > > We hope that our revisions and comments help the reviewer better assess Section 4, where we empirically validate our theory using our one-layer transformer model. The phenomenological theory is incomplete without these empirical tests, and we encourage the reviewer to revisit Sections 3 (+ the appendix) and 4.
> > > > > >
> > > > > > >Questions:
> > > > > > > What is the value used to plot Figure 4 and how does it affect the loss landscape?
> > > > > >
> > > > > > We set $z_{MLP} = 0$ in Figure 4, which is equivalent to setting $P^+(\phi) = \delta(\phi)$ (we have added a line in the caption to clarify). Note that Figure 4 shows the ICL loss landscape without the MLP.
> > > > > >
> > > > > > > Figure 5: the power law is presented as being between and but then Figure 6a presents it as being between and  How do these two representations relate?
> > > > > >
> > > > > > In Section 3 and Figure 5, we show empirically in MLP simulations that there is a power-law relationship between $K$ and $I_K(\infty)$. Our theory shows that this implies a power law relationship between $K^*$ and $N$, specifically, $K^* \sim N^{1/\nu}$ (Equation 14). In Figure 6a, we empirically verify this relationship using our one-layer transformer model. We think the power-law relation between $K^*$ and $N$ is the most non-trivial and stringent prediction of our theory, which is indeed validated in experiments (Figure 6a).

---

> > > > > > > ### Author Response · Authors · 2024-11-27
> > > > > > > **Final changes**
> > > > > > >
> > > > > > > This is a final comment to note that we significantly updated our main text to improve presentation. The detailed derivations are in the main text. Note that none of the results have changed.

---

> > > > > > > > ### Comment · Reviewer_EVew · 2024-12-03
> > > > > > > > **Final Comments to Authors**
> > > > > > > >
> > > > > > > > I thank the authors for their comments and revised draft of the paper. I acknowledge that some of my questions have been addressed in the comments, particularly revolving around the dimensionality of the input (although even in the rebuttal I believe the authors are not being clear enough on what is required for the model to work versus for the theoretical statements to hold - this is likely the source of my original concern that the authors are changing conditions to fit their point). I must say, that I disagree very strongly with the approach taken by the authors during this conference submission. Exemplified by the amount of content added to the appendix, it is clear that necessary detail had been omitted from the original paper. I reiterate my original sentiment that I suspect that the work is interesting - following the rebuttal I do still believe this to be the case. However, I find it highly inappropriate that the paper would be submitted in its original state and then 10 pages of very relevant and new appendix were added after the reviewing period.
> > > > > > > >
> > > > > > > > Thus, I will leave a comment here and trust the area chair will use my thoughts to their best discretion. In summary, I stand by my original assessment that the presentation of the theoretical details of the work (and lack thereof) meant the paper was not ready for publication. The appendix which has now been added does address this concern. I believe it is inappropriate to submit a paper without sufficient detail for initial review and then add the **entire derivation** after the review period as a rebuttal draft. I believe this sort of behaviour is why many conferences do not allow revision drafts. I leave the decision on whether this is appropriate for ICLR to the area chair. I commend the authors on the interesting work, but wish I had been given the opportunity by the authors to conduct a fair review from the beginning. I will leave my score the same but emphasise that it should be taken in the context of this statement.

---

> > > > > > > > > ### Author Response · Authors · 2024-12-03
> > > > > > > > >
> > > > > > > > > Thank you for your comment. We emphasize that the 5 pages of appendix are an expanded version of what was already in the main text, with certain derivation steps explicitly written out. No new results have been introduced and none of the previous results (except for the minor changes in the exponent noted in the main comment above) have changed.

---

### Official Review · Reviewer_6toB · 2024-11-04

**Soundness:** 3
**Presentation:** 3
**Contribution:** 3
**Rating:** 8
**Confidence:** 3

**Summary:**

The paper explores the mechanism by which models acquire the ability to generalize and memorize.
The authors test the hypothesis that generalization arises because, in models with limited capacity, a preference for generalization over memorization emerges when task diversity is sufficiently high.
The authors find that generalization is not solely the result of capacity constraints but also due to different learning kinetics between sub-circuits responsible for generalization and memorization.
At sufficient task diversity, the sub-circuits responsible for generalization can be acquired faster than those for memorization.
They also find that a network can lose generalization abilities in favor of memorization with sufficient training under $L_{2}$ regularization.
The paper identifies a number of properties about the dynamics of generalization and memorization.
Among them are how MLP-memorization scales with task diversity, how the task diversity threshold scales with context length, and the sensitivity of generalization to initialization.

**Strengths:**

The way models acquire generalization during training is an interesting topic and the paper provides an insightful exploration of the mechanisms involved.
The authors convincingly support the premise of the paper with theory and empirical studies.

**Weaknesses:**

Because the model used for the empirical validation is stripped down to a one layer transformer model the authors note that they cannot be sure that the dynamics are the same for much larger models.
The authors could consider supplementing the one-layer analysis with tests on a deeper transformer model.
Even if constrained by computational resources, experiments on a 2 or 3 layer transformer would help assess the applicability of their findings to larger networks.

The paper does not recommend methods to facilitate generalization. If possible, a discussion of how the ideas in the paper may be used to develop initialization and regularization techniques would make the submission even stronger.

**Questions:**

Can the authors use this work to recommend methods to improve generalization? For example, can we choose an initialization method that might better support ICL?

The theory hints that ICL decays due to the use of regularization. Though using $L_2$ regularization makes the results more comparable with typical training, would it be possible to provide empirical results without regularization?

---

> ### Comment · Reviewer_6toB · 2024-11-26
>
> I thank the authors for their work during the discussion phase. I have been following the discussion initiated by reviewer EVew. I remain of the view that the submission is strong and will retain my score of 8.

---

> ### Author Response · Authors · 2024-11-27
> **Responses to Reviewer 6toB**
>
> We thank the reviewer for their review of our article. Here are our responses to their comments:
> > The authors could consider supplementing the one-layer analysis with tests on a deeper transformer model. Even if constrained by computational resources, experiments on a 2 or 3 layer transformer would help assess the applicability of their findings to larger networks.
>
> We provide additional experimental results using a 2-layer transformer model (consisting of two blocks of attention followed by the 3-layer MLP) in supplementary figure A5. We replicate our finding that the time to acquire ICL increases linearly with context length, in the limit of large data diversity. We also observe that the distribution of the time taken to acquire ICL is also broadly long-tailed.
>
> > Can the authors use this work to recommend methods to improve generalization? For example, can we choose an initialization method that might better support ICL?
>
> Our work suggests that context length is an important consideration in ICL, as the median time to acquire ICL can increase linearly with context length. While we do not have specific recommendations, we show that the memorization kinetics of the MLP (characterized by the time evolution of the MLP order parameters) can influence when ICL is acquired. This dovetails with our explanation for how regularization can cause ICL to decay, explaining the observations from Singh et al. that applying weight decay only to self-attention layers cause ICL to be transient, whereas applying weight decay only to MLP layers mitigates ICL transience.
>
> > The theory hints that ICL decays due to the use of regularization. Though using $L_2$ regularization makes the results more comparable with typical training, would it be possible to provide empirical results without regularization?
>
> We provide additional experimental results in Figure A6 showing how ICL performance varies as we tune regularization of the self-attention layer. We show that when regularization is too low, we do not observe ICL transience as the time to observe transience is too long. When regularization is too high (above $4\times10^{-3}$), the model never attains the ICL solution. In intermediate values of regularization, the model does attain transience in a reasonable amount of time. In this regime, we observe that increasing regularization induces faster transience.
>
> We hope that these additional experiments will help bolster our case that our theoretical results apply to larger networks and clarify how regularization affects ICL transience.

---

### Author Response · Authors · 2024-11-25
**Revised version of the manuscript with a much expanded Appendix**

A new version of the manuscript has been uploaded where we significantly expand on the analysis of the minimal model in the appendix. We hope that this new version will address the general concern raised by Reviewer EVew. We are working to further revise our manuscript based on all reviewer comments and plan to respond to each review in further detail over the next few days.

---

### Author Response · Authors · 2024-11-27
**Revised manuscript**

We thank all the reviewers for the detailed comments -- we think the manuscript has improved significantly after incorporating feedback.

The major changes are summarized below:
1) Added an Appendix detailing the mathematical derivations.
2) The Appendix allowed us to cut down on derivations in the main text and use that space to expand on the intuition and conceptual ideas.
3) Three new/updated figures: New simulations with a two-layer model (Supp Fig A.5), exploring how regularization impacts transience (Supp Fig A.6) and an updated Supp Fig A.3 with 10x more iterations.

Note that none of the results have changed except for a small change in the measured value of $\nu$ (from 0.72 to 0.70) to reflect the updated Supp Fig A.3 with 10x more iterations.

---

### Meta-Review · Area_Chair_ToDw · 2024-12-20

**Metareview:**

### Summary

The paper proposes a simple framework to understand the emergence of In-Context Learning (ICL).
The framework shows the presence of a competition between ICL and In-Weight Learning.
The analysis, based on a series of ansatz, allowed the authors to characterise the dynamics of the two types of learning and understand why in some cases ICL may not emerge or may display a transient nature.

### Strength

The model is novel and interesting.
While still consisting in a single-layer attention network, it goes beyond previously introduced models and captures and explains a wide range of phenomena.

### Weakness

There are still clear limitations in the theory.
The architecture considered is far from applications and data structure is extremely simplified.

### Reason for acceptance

The paper represents a step forward in the understanding of transformers and provides a valuable contribution for the conference.

**Additional Comments On Reviewer Discussion:**

One of the main critiques in the discussion period was about the **extremely high-level description of the analysis**.
I support this concern because, as Reviewer EVew reported, this can favour the proliferation of erroneous results.
I believe that the first submission was at a very preliminary stage and by the end of the discussion the paper showed a great improvement in terms of clarity. In particular, the appendix changed from being a quite chaotic collection of figures to an actual explanation of the thought process. Nevertheless, after comparing the first and last versions, I believe that the key ideas for the derivation were already present and I am recommending acceptance.

The other concerns were related to **clarity** and **connections to realistic models**. They were addressed by the authors in the discussion and the reviewers were satisfied as shown by their gradings.

---

### Decision · Program_Chairs · 2025-01-22

Accept (Spotlight)